# Gephyrin filaments represent the molecular basis of inhibitory postsynaptic densities

Arthur Macha[1], Filip Liebsch [ID][1], Emanuel H. W. Bruckisch [ID][1], Nele Burdina[1], Imke von Stülpnagel [ID][1], Konrad Benting [ID][1], Monika Gunkel [ID][1], Elmar Behrmann [ID][1] [✉] & Guenter Schwarz [ID][1,2] [✉]

The multifunctional protein gephyrin clusters inhibitory receptors at the postsynaptic membrane in the CNS. Gephyrin has been proposed to form the inhibitory postsynaptic density by liquid-liquid phase separation, involving a complex interplay between receptor binding and oligomerization via its conserved G- and E-domains. Here we show by single particle cryo-EM analysis that dimerization promotes the formation of gephyrin filaments in which two E-domain dimers are linked by Z-shaped interfaces formed between two subdomains II (SDII) of adjacent dimers. Deletion of SDII, introduction of two epilepsy-causing pathogenic variants, or neutralization of an opposing charge in the interface abolish the formation of filaments, in vitro phase separation, and synaptic receptor clustering in hippocampal neurons. In conclusion, this work identifies gephyrin E-domain filaments as the structural foundation underlying gephyrin both phase separation and receptor clustering at inhibitory postsynaptic densities.

Gephyrin (Geph) is the major scaffold protein at inhibitory synapses in the central nervous system (CNS)[1–3]. It directly interacts with the intracellular domains (ICDs) of the β-subunit of glycine receptors (GlyR) as well as with various subunits of γ-aminobutyric acid type A receptors (GABA_ARs)[4–8]. These interactions facilitate the incorporation of inhibitory receptors into the postsynaptic density (PSD), which involves many other interacting proteins such as tubulin, collybistin and neuroligin[9–13]. The presence of a sufficient number of inhibitory receptors at the synaptic site is essential for efficient inhibitory signal transmission between neurons[14,15].

Gephyrin was initially identified as a 93 kDa protein associated with GlyR[16–18]. It is comprised of an N-terminal G-domain (GephG), an E-domain (GephE), and a flexible central linker region, referred to as the C-domain (GephC) (Fig. 1a)[4,19,20]. Structural analyses showed that isolated GephG forms trimers[19,21], while GephE forms dimers[4,20,22]. Biochemical studies suggest that full-length Geph is a trimer, mediated by the central GephG trimer[9,16,20]. The interaction with the ICDs of inhibitory receptors occurs at the dimerized GephE, in which the two GephE subunits form two C-shaped hydrophobic binding pockets[23,24]. In vitro interaction studies using gephyrin trimer and a peptide derived

from the ICD of the GlyR-β subunit (GlyR-loop) revealed biphasic binding kinetics, characterized by distinct high-affinity and low-affinity interactions with dissociation constants ($K_D$) in the low nanomolar to low micromolar range[4,20]. The ICDs of GABA_ARα1, GABA_ARα3, GABA_ARβ2, and GABA_ARβ3 subunits bind to the same binding pocket as the GlyR-loop, but with approximately 20–50 times lower affinity[5,7].

Based on the distinct oligomerization properties of the G- and E-domains of gephyrin, a hexagonal lattice has been proposed as the structural basis for the inhibitory postsynaptic density[22,25]. While such an ordered lattice has neither been observed in vitro nor in vivo, a complex network of receptors at the synaptic cleft has been identified by cryo-electron tomography that appears to be supported by a thin electron-dense layer that would agree with a network of Geph[26]. Furthermore, structural and functional data revealed that localized variations in protein density of both receptors and scaffolding proteins can lead to the formation of highly stable nanodomains within the PSD[27–30]. On a conceptual level, liquid–liquid phase separation (LLPS) has been proposed as an underlying principle of gephyrin clustering and inhibitory synapse formation and linked to the unstructured and charged GephC domain based on in vitro assays[31,32]. However, the

[1]Institute of Biochemistry, Department of Chemistry and Biochemistry, University of Cologne, Cologne, Germany. [2]Center for Molecular Medicine Cologne (CMMC), University of Cologne, Cologne, Germany. [✉]e-mail: ebehrman@uni-koeln.de; gschwarz@uni-koeln.de

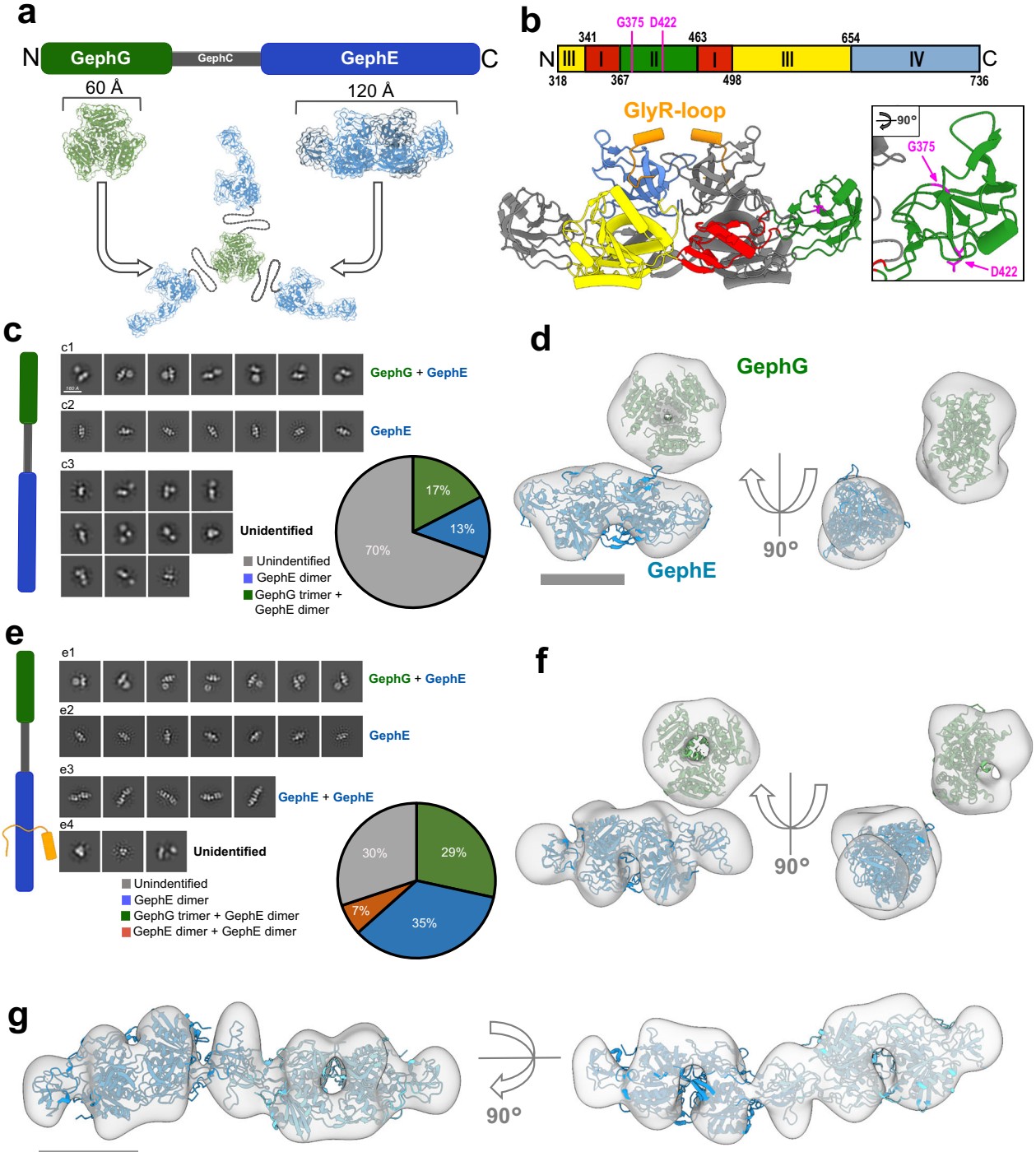

**Fig. 1 | GlyR-loop stabilizes GephE dimerization within gephyrin trimer.**
**a** Gephyrin trimer domain structure overview showing G-domain (GephG), C-domain (GephC) and E-domain (GephE). The respective crystal structures of GephG (green, PDB: 1JLJ) and GephE (blue, PDB 2FU3) are shown in cartoon representation with faded surfaces and size indicators. A graphical representation of the full-length gephyrin trimer is depicted (middle panel). **b** Cartoon representation of isolated GephE dimer in complex with GlyR-loop (PDB: 2FTS), highlighting subdomain (SD) division (color code: SDI: red; SDII: green; SDIII: yellow; SDIV: blue) and position of GlyR-loop (orange). The second monomer is shown in gray. Positions of G375 and D422 in SDII are highlighted (magenta). **c** Representative 2D class averages of isolated Geph trimer following negative stain SPA showing GephE with GephG (c1), isolated GephE (c2) and undefinable densities (c3). The pie chart indicates the relative distribution of particles throughout the 2D class averages. **d** Surface representation of final map using particles from classes with GephE + GephG (c1). GephE dimer (PDB 2FU3, blue) and GephG trimer (PDB: 1JLJ, green) shown in cartoon representation were rigid-body fitted into the respective densities in the map. Scale bar: 50 Å. **e** Representative 2D class averages of isolated Geph trimer in complex with GlyR-loop after negative stain SPA showing GephE with GephG (e1), isolated GephE (e2), complex of two GephE dimers (e3) and undefinable densities (e4). The pie chart indicates the relative distribution of particles throughout the 2D class averages. **f** Surface representation of final map using particles from classes with GephE + GephG (e1). GephE dimer (PDB 2FU3, blue) and GephG trimer (PDB: 1JLJ, green) shown in cartoon representation were rigid-body fitted into the respective densities in the map. **g** Surface representation of final map using particles from classes with GephE + GephE complex (e3). Two consecutive GephE dimers shown in cartoon representation (PDB 2FU3, blue) were rigid body fitted into the respective densities in the map. Scale bar 50 Å. Cartoon bar diagrams illustrate the Geph constructs used in this experiment (green: GephG, gray: GephC, light green: SDII, red: position of amino acid substitution, orange: GlyR-loop).

molecular mechanism that enables Geph to self-assemble at the PSD remained elusive for nearly three decades, and neither the hexagonal lattice nor the LLPS hypothesis have been able to fully explain known pathogenic variants of Geph, as highlighted by Geph[G375D] and Geph[D422N] [14,33], which harbor mutations outside the known receptor and oligomerization sites (Fig. 1b). In addition to its function in the CNS and spinal cord, Geph also harbors an enzymatic function in molybdenum cofactor biosynthesis[34–36].

Here we elucidated a hitherto unknown molecular interface formed between subregions of GephE, which not only rationalizes two known pathogenic patient variants carrying point mutations in this subregion, but also allowed the rational design of a third variant again disrupting Geph oligomerization. Our negative stain EM data, in combination with functional data, revealed that the presence of the GlyR-loop stabilized intramolecular GephE dimerization within full-length Geph. Subsequent cryo-EM data of the Geph-GlyR-loop complex uncovered that Geph is able to form flexible filaments, in which dimerized GephE domains are connected via their subdomain II (SDII). Geph variants either lacking SDII or carrying known pathogenic amino acid exchanges consistently lose filament-forming capability, are unable to engage in LLPS in vitro, and are deficient in synapse formation in vivo. Moreover, the structural data allowed predicting a complementary third variant showing the same kind of loss in filament formation and phase separation. Collectively, our structural and functional studies highlight Geph filament formation as an underlying molecular mechanism of Geph-mediated postsynaptic clustering of inhibitory neurotransmitter receptors.

## Results

### GlyR-loop stabilizes GephE dimerization within full-length Geph

Given that the full-length structure of gephyrin remains unknown, we carried out single-particle analyses (SPA) of isolated wild-type gephyrin trimers (Geph[WT]). For this, Geph[WT] was overexpressed in *E. coli* and purified to homogeneity using affinity chromatography combined with analytical scale size-exclusion chromatography (aSEC) (Supplementary Fig. 1a), and then prepared for negative stain EM. Micrographs showed particles of varying sizes (Supplementary Fig. 2a) with a high degree of heterogeneity; however, common architectural principles could be identified after 2D classification (Fig. 1a, c). A subset of class averages, accounting for about 17% of particles, showed an elongated bilobed structure adjacent to a round structure. Based on the molecular dimensions of the GephG (PDB 1JLJ) and GephE (PDB 1JLJ) crystal structures, these shapes likely represent a GephG trimer and GephE dimer, respectively (Fig 1c1). A second subset of class averages, accounting for about 13% of particles, showed a single elongated bilobed structure, likely representing a GephE dimer without a resolved GephG trimer (Fig 1c2). A third subset of class averages showed structures of varying shapes and sizes that could not be confidently assigned to any gephyrin domain (Fig 1c3). Approximately 70% of all particles fell into this heterogeneous group of class averages, suggesting that the majority of gephyrin trimers adopted heterogeneous conformations, consistent with structural flexibility observed in previous SAXS and AFM studies[20]. The particle subset showing putative GephG and GephE densities in close proximity (Fig 1c1, green) was further processed in 3D (Supplementary Fig. 2b), yielding an intermediate-resolution negative stain EM density map that features two distinct, disconnected densities that were confirmed as GephE and GephG by rigid-body fitting known crystal structures (PDB 2FU3 and PDB 1JLJ) indicating the presence of stable GephE dimers and GephG trimers in our sample (Fig. 1d and Supplementary Fig. 2c). Neither density corresponding to the third copy of GephE nor density that might correspond to GephC could be identified.

We next investigated if the addition of GlyR-loop, as a mimic for receptor ICD binding, affects the architecture of full-length Geph[WT]. For this we added a 3-fold excess of GlyR-loop to Geph[WT] and subsequently isolated the complex by aSEC before staining for negative stain EM (Supplementary Figs. 1a and 3). Micrographs appeared similar to those obtained for the sample without the GlyR-loop (apo-Geph), showing particles of varying sizes (Supplementary Fig. 3a), which, after processing resulted in visually similar 2D class averages as before (Fig. 1e). However, in contrast to apo-Geph, the relative number of particles contributing to the heterogeneous group of class averages was decreased more than two-fold (Fig 1e1-4), implying that the presence of the GlyR-loop reduced heterogeneity by stabilizing Geph in defined conformations. Again, the particle subset showing GephG and GephE in close proximity (Fig 1e1) was further processed in 3D, revealing a structure composed of two distinct densities that was similar to the one obtained for apo-Geph (Fig. 1f and Supplementary Fig. 3b). However, while nominal at the same resolution and only 50% more particles contributed to the reconstruction, the structure was much better defined, particularly in the GephE region (Fig. 1d, f). Still, stain granularity prevented the direct observation of GlyR-loop binding.

Interestingly, 2D classification revealed an additional subset of 2D class averages that appeared to comprise linear assemblies of two or more GephE dimers (Fig 1e3). We also processed this particle subset showing linear assemblies (Fig 1e3) in 3D, resulting in an elongated negative stain EM density that allowed rigid-body fitting of two GephE dimers, which appeared to be linked via their respective SDIIs (Fig. 1g and Supplementary Fig. 3c).

In conclusion, our SPA analysis of full-length Geph revealed high structural heterogeneity, which was decreased following the incubation with GlyR-loop. The presence of GlyR-loop, moreover, triggered the formation of linear assemblies that appear to comprise two or more directly interacting GephE dimers.

### Full-length Geph forms flexible filaments at high protein concentrations

To uncover the molecular basis for the linear assemblies of Geph observed in negative stain EM, we investigated full-length Geph[WT] trimers complexed with the GlyR-loop by cryo-EM. Under identical sample preparation conditions as used for negative stain EM, except at a 100-fold increased protein concentration, we obtained micrographs that were densely packed with particles (Supplementary Fig. 4a). Subsequent unsupervised 2D classification resulted in a set of high-quality 2D class averages featuring distinct secondary structure elements (Fig. 2a). While some 2D class averages were centered on the GephE dimer, others were centered on a Z-shaped structural feature connecting two GephE dimers (Fig. 2a and Supplementary Fig. 4b). 2D class averages centered on the GephE dimer showed weak adjacent density, indicating that these GephE dimers are likely incorporated into larger linear assemblies, or filaments, in agreement with our negative stain EM data (Figs. 1e3 and 2a). Therefore, we re-extracted these particles using a box size twice as big as before and repeated the unsupervised 2D classification. The resulting 2D class averages confirmed the presence of slightly curved GephE filaments comprising GephE dimers interconnected by a Z-shaped interface (Fig. 2a and Supplementary Fig. 4c). We additionally observed 2D class averages showing GephE dimers in close proximity to a smaller bilobed density, which could be identified as a side view of the GephG trimer based on in silico 2D back-projections of the GephG crystal structure (Fig. 2a, b). However, subsequent 3D reconstruction of the GephE filament failed due to the crowdedness of these grids and a high degree of preferential orientation of the particles (Supplementary Fig. 4d).

Given that Geph complexed with the GlyR-loop eluted in SEC homogeneously at the expected elution volume of the trimeric Geph-GlyR complex (Supplementary Fig. 1a), we concluded that the observed GephE filaments must have been formed during the cryo-EM sample grid preparation. As the vitrification process itself has been shown to only minimally affect the sample, especially with regard to

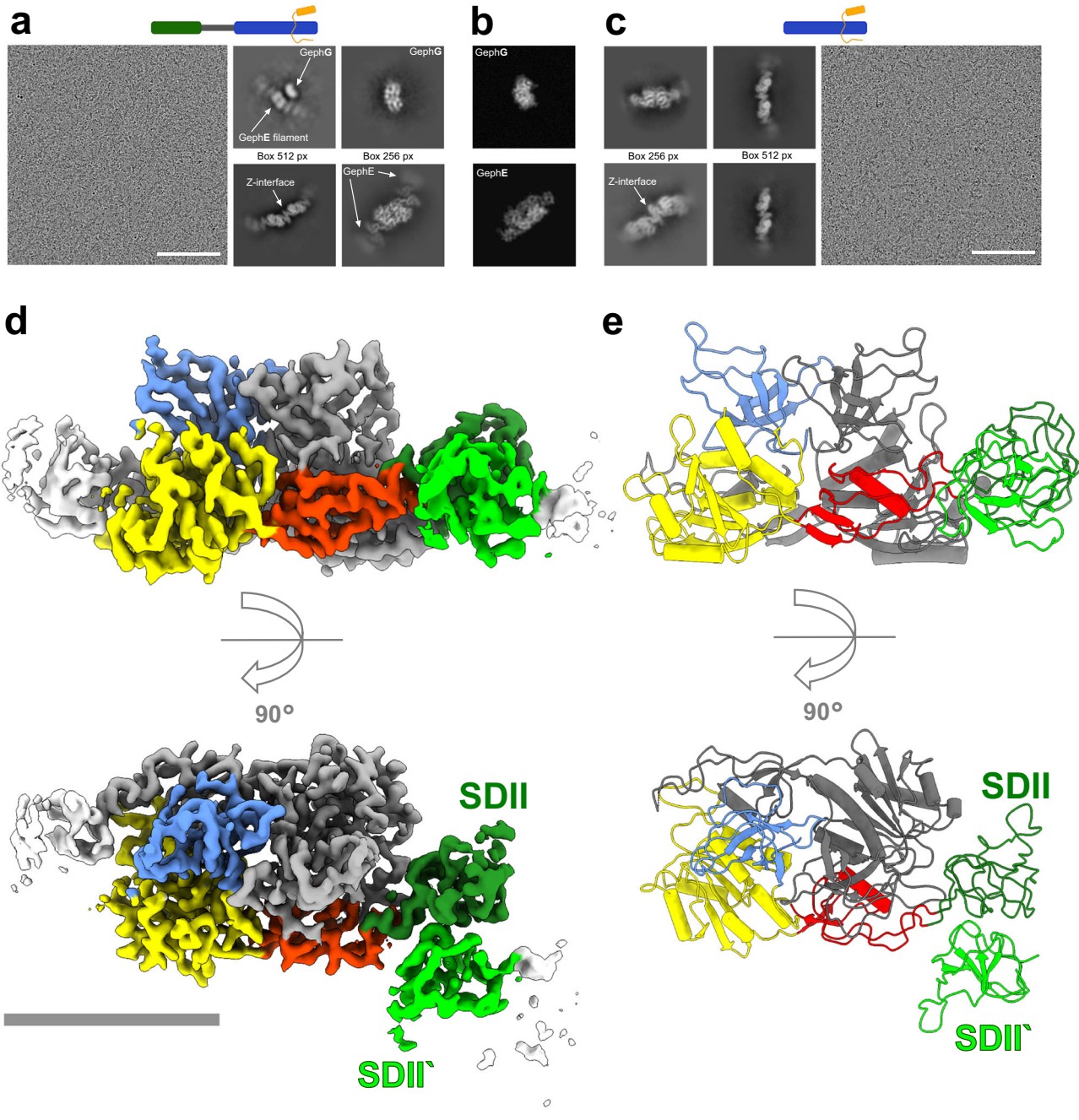

**Fig. 2 | Cryo-EM SPA of gephyrin reveals formation of filaments facilitated by GephE SDII interactions. a** Representative micrograph (scale bar 100 nm) of Geph trimer in complex with GlyR-loop with representative 2D class averages depicting GephG and GephE incorporated into filaments through a Z-shaped interface (arrow). Densities corresponding to GephE and GephG are highlighted in the 2D class average pictures, respectively (arrow). **b** Representative 2D back-projections of GephG (PDB: 1JLJ) and GephE (PDB 2FU3). **c** Representative micrograph (scale bar 100 nm) with GephE in complex with GlyR-loop with representative 2D class averages depicting GephE incorporated into filaments through a Z-shaped interface (arrow). **d** Surface representation of cryo-EM map of GephE dimer inserted into filaments with highlighted SDI (red), SDII (dark green), SDIII (yellow), SDIV (blue) from one GephE monomer and SDII from adjacent GephE dimer (SDII', light green). The second GephE monomer is colored in gray. Regions of the structure not used for model building are shown in white. Scale bar 50 Å. **e** Final structure of GephE, shown on a cartoon representation incorporated into filaments, is built into the map shown in (**d**), highlighting the SDs positions and the SDII-SDII' interface. Cartoon bar diagrams illustrate the Geph constructs used in this experiment (green: GephG, gray: GephC, light green: SDII, red: position of amino acid substitution, orange: GlyR-loop).

processes on the time-scale of protein oligomerisation[37], we hypothesize that the blotting step might have induced the oligomerisation process, possibly due to a rapid increase in local protein concentration as observed for other proteins[38].

Because in our negative staining SPA elongated assemblies of GephE dimers were only observed when Geph was complexed with the GlyR-loop, but not for apo-Geph, we also prepared cryo-EM samples in the absence of the GlyR-loop. Resulting micrographs were of comparable quality, also showing densely packed particles (Supplementary Fig. 5a). Interestingly, despite the absence of the GlyR-loop, data processing revealed 2D class averages that were similar to those observed in the presence of the GlyR-loop, showing GephE filaments alone or in

close proximity to a smaller bilobed density (Supplementary Fig. 5b), implicating that potentially the increased concentration might overcome the necessety for GlyR-loop binding observed during negative stain EM.

## GephE filaments are connected via an interlocking interface of neighboring SDIIs

In order to confirm that neither GephG nor the C-linker is involved in the formation of the filaments, and aiming to reduce the crowdedness of the sample on the grid, we prepared cryo-EM samples of isolated GephE (residues 327–736) using identical experimental conditions as for full-length Geph before (Supplementary Fig. 6). Again, 2D class averages revealed linear, but flexible assemblies of GephE dimers connected by a Z-shaped interface (Fig. 2c and Supplementary Fig. 6c).

To mitigate the effects of preferred orientation, we acquired an additional dataset at a 25° tilt of the sample stage (Supplementary Fig. 6b). In this dataset, several micrographs showed highly reduced particle crowdedness and clearly illustrated the distribution and proportions of filaments on the micrographs (Supplementary Fig. 6b1). Processing of combined datasets yielded extended particle orientations (Supplementary Fig. 6d), which allowed for an ab-initio 3D reconstruction that revealed a complete GephE dimer with additional density on one side, corresponding to the SDII of an adjacent GephE dimer in the filament (hereafter referred to as SDII') (Supplementary Fig. 6e). Further processing yielded a final map at a global resolution of 3.5 Å, showing clear side-chain density in most areas (Supplementary Fig. 6f). However, density for the SDII-SDII' interface remained fragmented, which we attributed to conformational flexibility in agreement with the various degrees of curvature observed for the 2D class averages of the filament (Fig. 2c and Supplementary Fig. 6c, d). Using a motion-based deep generative model for continuous heterogeneity[39], the flexibility of SDII and the SDII-SDII' interface to the adjacent GephE dimer was confirmed (Supplementary Fig. 6g).

In order to improve the resolution in the SDII-SDII' interface region, we used in silico-generated 3D structures based on the deformation flow field from the flexibility analysis to initiate a 3D classification. This yielded one class with a poorly resolved SDII-SDII' interface (35% of particles) and two classes with improved resolution at the SDII-SDII' interface (65% of particles, Supplementary Fig. 6h), of which the latter subset of particles was further refined. Global refinement and subsequent local refinement focussing on the area encompassing the core part of GephE dimer including the SDII-SDII' interface yielded a map with continuous density at the SDII-SDII' interface and a resolution of approximately 3.0 Å in the core part of the GephE dimer and 3.5 to 4.0 Å in the SDII-SDII' interface (Fig. 2d; Supplementary Figs. 6i, j and 7a), allowing us to build an almost complete atomic model into the density map following additional deep-learning based[40] map sharpening (Fig. 2e and Supplementary Fig. 6j). In our atomic model chain A comprises all four subdomains of GephE (Fig. 2e). Due to the highly fragmented density of chain B in the SDII region, the respective amino acid residues (Ala370–Pro459) were not build into our atomic model (Fig. 2e). For SDII' (chain C) we utilized a predicted structure of SDII as a starting template for building the atomic model[41,42].

Overall, our structure of the GephE dimer highly agrees with previous crystal structures, especially in the regions of SDI, SDIII and SDIV[4]. The here identified SDII-SDII' interface features a prominent groove on the SDII side that interlocked with a complementary bulge on the SDII' side (Fig. 2e and Supplementary Fig. 7b, c).

## Plasticity of the SDII-SDII' interface supports GephE filaments flexibility

While our 3D sorting strategy could alleviate the fragmentation of the SDII-SDII' interface density, likely caused by hinge-like motions at the two loops connecting SDI and SDII, the lower resolution of this area in our final reconstruction implied the presence of remaining structural flexibility. To further characterize potential heterogeneity of the SDII-SDII' interface we re-processed the particles from the initial 3D sorting (Supplementary Figs. 6h and 8). For this, particles that did not show a well-resolved SDII-SDII' interface region ("fragmented"), and particles that showed a well-resolved SDII-SDII' interface region ("defined"), were individually refined and then subjected to a motion-based deep generative model for continuous heterogeneity in order to visualize the main modes of SDII-SDII' interface movement (Supplementary Fig. 8a, b). Notably, the consensus position of the SDII' in the structure from the "fragmented" subset was shifted by approximately. 9.8° upward as compared to the previously identified, higher resolved, SDII' position (Fig. 3a and Supplementary Fig. 8c). SDII and SDII' from the "defined" subset showed a movement of up to 3.9° and 9.5° for SDII and SDII', respectively (Fig. 3b and Supplementary Fig. 9a). In contrast, the movement of the interface with the uptilted SDII' showed an increased movement of up to 9.9° for SDII and 20.9° for SDII' (Fig. 3c and Supplementary Fig. 9b). Potentially, the later "fragmented" particle population (Supplementary Fig. 8a) might represent a less stable state before SDII and SDII' fully interlock.

To follow up on the observed mobility of the SDII, we investigated whether the incorporation of GephE into filaments changes the position of the SDII in relation to the other domains of the protein. When comparing SDII positions in the crystal structure of human GephE either without or with bound GlyR-loop (PDB: 2FU3, 2FTS) we found that SDII in our structure is positioned further away from the core part of the protein compared to the GephE crystal structure without bound GlyR-loop (Fig. 3d). This difference was smaller if comparing to the GlyR-loop bound GephE crystal structure, implying a GlyR-loop dependent shift in the SDII positioning might be related to filament formation as it promotes a positioning of the SDII more similar to the positioning observed in GephE filaments (Fig. 3e and Supplementary Fig. 9d). A similar effect was observed in the crystal structures of the GephE plant orthologue, Cnx1E, that shows a shift of SDII towards the filament-promotion angle when the cofactor was bound (Fig. 3f, g and Supplementary Fig. 9e, f)[43,44].

## A single GlyR-loop is able to occupy both binding sites within one GephE dimer

Despite saturation of the sample with the GlyR-loop, the consensus structure showed only fragmented density in the receptor binding sites (Figs. 1b and 2d). We hypothesized that this might be caused by the GlyR-loop adopting multiple conformations within the binding pocket[4,45]. In order to better resolve the GlyR-loop density, we carried out a 3D classification of all particles of GephE irrespective their SDII conformation using a mask focusing on the core part of the GephE dimer, including both receptor-binding sites (Supplementary Fig. 6f; Sub-Fig. 10). The subset of particles showing the most continuous density for the GlyR-loop was further processed by local refinement excluding the SDII subdomains (Supplementary Fig. 10a). This yielded a final map that after low-pass filtration to 6 Å showed a continuous density spanning both binding pockets on the GephE dimer (Fig. 4a; Supplementary Fig. 10a). 3D classification resulted in classes that either showed GlyR-loop density in both binding sites or a complete absence of GlyR-loop density, suggesting that in the case of binding, a single GlyR-loop is simultaneously occupying both binding sites within one GephE dimer (Fig. 4a, b; Supplementary Fig. 10b). Importantly, this finding rationalizes previous biochemical data reporting the binding of one GlyR-loop to one GephE dimer[4,23,31,45]. Moreover, machine-learning-based structure predictions[46] of the most probable position of a single GlyR-loop within the GephE dimer are in line with the observed density for GlyR-loop in our reconstruction (Fig. 4c–e and Supplementary Fig. 11), and predict an extension of the GlyR-loop core binding motive (R394-Y412) towards the C-terminal end, enabling the occupation of both binding sites simultaneously (D413-L426, Fig. 4d, e). This agrees

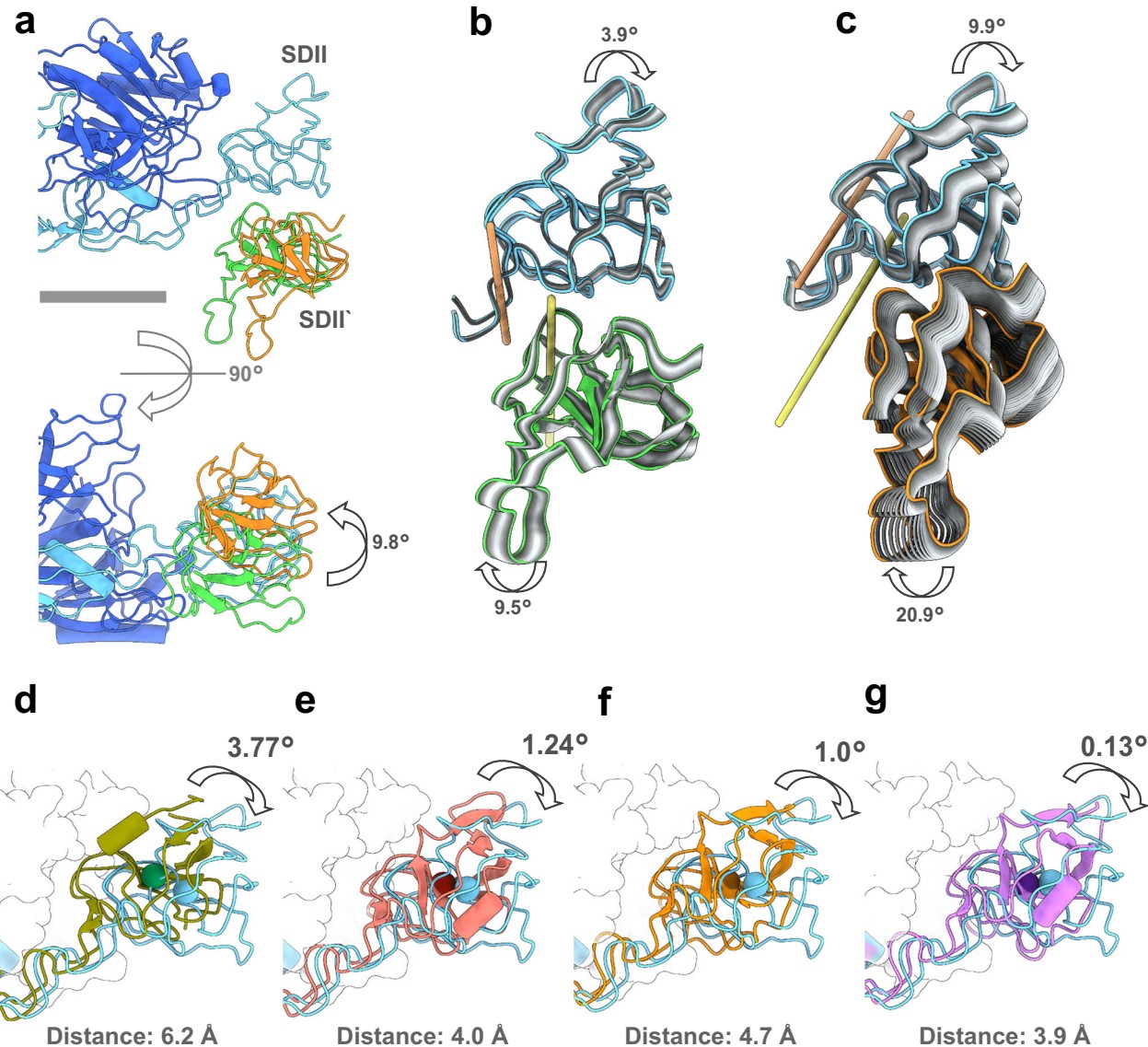

**Fig. 3 | The SDII-SDII′ interface allows for motion between GephE dimers.**
**a** Overlay of the alternative SDII′ binding position (orange) within the SDII-SDII′ interface with the main position observed for SDII′ in the interface (green). For orientation, SDI and SDII of one GephE subunit (light blue), and SDII and SDIV of the second GephE monomer (dark blue) in the dimer are shown. The shift between the SDII′ positions is indicated. For further information, see Supplementary Fig. 8. Scale bar is 25 Å. **b**, **c** Visualization of main trajectories depicting possible degrees of freedom of SDII (blue) and SDII′ (green or orange) in the main (**b**) and alternative (**c**) binding positions based on motion-based deep generative modeling. Strongest deviation from the consensus position is highlighted in color, the course of movement shown in shades of gray, and the difference between the extreme positions indicated in degrees. For further information see Supplementary Figs. 8 and 9d–g) Structural comparison of the SDII position in the GephE dimer incorporated into filaments in the presence of GlyR-loop (blue) with the crystal structure without GlyR-loop (PDB: 2FU3, olive (**d**), the GephE crystal structure bound to GlyR-loop (PDB: 2FTS, red (**e**), the Cnx1E crystal structure without Moco-AMP (PDB: 5G2R, dark orange (**f**), and the Cnx1E crystal structure bound to Moco-AMP (PDB: 6Q32, purple (**g**). Differences in SDIIs positions are indicated in distances and angular changes. For further information, see Supplementary Fig. 9.

with previous in vitro studies showing that residues 413-426 are required for high-affinity binding of the GlyR-loop to GephE dimer[23].

## SDII is crucial for Geph filament formation, LLPS, and clustering at synapses

To confirm that the observed filament formation is solely dependent on SDII interactions, we generated a full-length Geph variant lacking the SDII domain (Geph$^{\Delta SDII}$, deleted residues 371–459) and purified the protein in complex with GlyR-loop to homogeneity similarly to the wild-type protein previously. Structural analysis revealed dimeric and trimeric particles, but no signs of filament formation as seen with wild-type protein, confirming that also in the presence of GephC and GephG filament formation requires SDII (Supplementary Fig. 12). Importantly,

binding studies with GlyR-loop showed that both full-length Geph$^{\Delta SDII}$ as well as GephE$^{\Delta SDII}$ maintained similar binding affinities as compared to the wild-type protein, indicating that the deletion of SDII does not interfere with receptor binding and GephE dimerization (Supplementary Fig. 1c–f).

Next, we asked whether the deletion of SDII would abolish Geph LLPS in vitro, which has been hypothesized to represent the molecular basis of Geph clustering[31,32,47]. For this, we mixed the GlyR- and GABA$_A$R-mimetic ICD proteins GlyR-LD, GABA$_A$R-α1LD, and GABA$_A$R-α3LD as described previously[31] with GephE variants and performed sedimentation experiments. In the presence of GlyR-LD, approx. 10 times more GephE$^{WT}$ was detected in the pellet fraction as compared to the sample without GlyR-LD, thus demonstrating strong

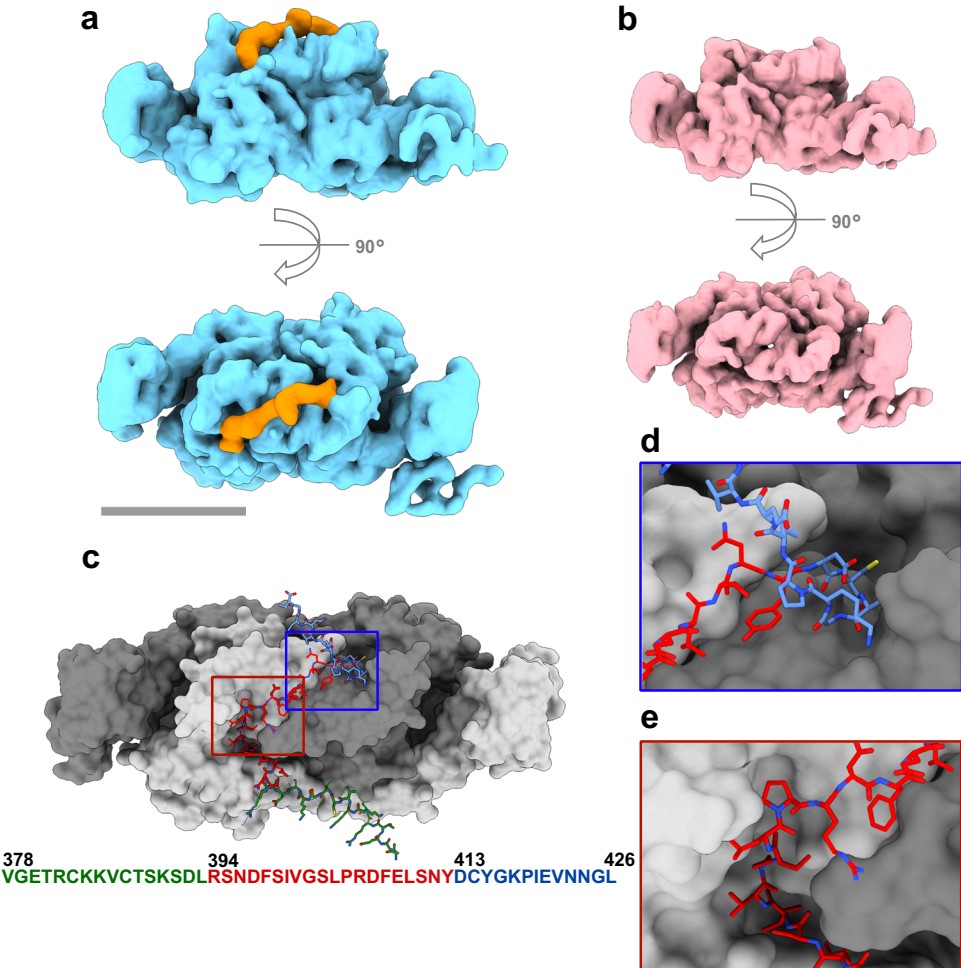

**Fig. 4 | GlyR-loop occupies both binding sites within one GephE dimer simultaneously. a** Surface representation of GephE dimer in complex with GlyR-loop lowpass filtered to 6 Å (light blue) with highlighted density depicting the position of GlyR-loop (orange). **b** Surface representation of GephE dimer without GlyR-loop lowpass filtered to 6 Å. For further information, see Supplementary Fig. 10). **c** GephE dimer in complex with GlyR-loop modeled after structure prediction. GephE dimer is shown in surface presentation with highlighted GephE monomers (light gray, dark gray). GlyR-loop is shown in stick presentation with color-coded amino acid sequence. Amino acid sequence of modeled GlyR-loop with respective positions is depicted beneath. **d** Magnification of GlyR-loop binding site one occupied by the core binding motive of GlyR-loop (red). **e** Magnification of second GlyR-loop binding site occupied by the C-terminally extended binding motive of GlyR-loop (blue).

LLPS formation (Fig. 5a, b). In contrast, Geph$^{\Delta SDII}$ did not sediment in the presence of GlyR-LD, as the band in the pellet fraction was similar to the control without GlyR-LD (Fig. 5a, b). Similar results were obtained for GABA$_A$R-α1LD and GABA$_A$R-α3LD, with again the amount of Geph$^{\Delta SDII}$ within the pellet fraction being unaffected by the presence or absence of the respective receptor ICD model proteins (Supplementary Fig. 1g, h). This data collectively demonstrated that SDII is required for Geph LLPS.

To confirm the physiological relevance of SDII-mediated Geph filament formation for postsynaptic Geph clustering, we expressed mScarlet-tagged gephyrin constructs in dissociated murine hippocampal neurons lacking endogenous gephyrin. This was achieved by infecting hippocampal cultures from floxed gephyrin mice[48] with recombinant adeno-associated viruses (rAAVs) encoding Cre recombinase and subsequently with rAAVs encoding Cre-dependent mScarlet-Geph or mScarlet-Geph$^{\Delta SDII}$, before the onset of synapse formation[49,50]. After 14 days in vitro, mScarlet-Geph formed frequent somatic and dendritic clusters, which co-localized with pre-synaptic vesicular GABA transporter (vGAT) and post-synaptic GABA$_A$Rs (GABA$_A$R-γ2), both markers of GABAergic synapses. In contrast, mScarlet-Geph$^{\Delta SDII}$ was diffusely distributed and did not form postsynaptic clusters (Fig. 5c, d), despite the fact that Geph$^{\Delta SDII}$ retained its

receptor-binding but lost its filament formation ability. Overall, these experiments demonstrate that the SDII subdomain-mediated filaments are a prerequisite for the formation of postsynaptic clusters at GABAergic synapses.

### Loss of Geph filaments provides the molecular basis of epileptic encephalopathy in two patients

After showing that SDII is required for Geph filament formation, LLPS and receptor clustering, we asked whether we can disclose the mechanistic basis of epileptic encephalopathy in previously identified pathogenic variants of Geph, Geph$^{G375D}$ and Geph$^{D422N}$, which are both located within SDII and in case of Geph$^{G375D}$ shows a strong impairment in LLPS formation[14,31,33].

Similar to GephE$^{WT}$, we purified GephE$^{G375D}$ to high homogeneity and added the GlyR-loop in a five-fold molar excess before cryo-EM sample preparation (Supplementary Fig. 1b). The resulting micrographs were of high quality, showing numerous evenly distributed particles (Supplementary Fig. 13a). Unsupervised 2D classification yielded high-quality class averages showing dimeric structures from various angles, but no indications of filament formation. Moreover, the observed structures were more compact than those observed for GephE$^{WT}$ (Supplementary Fig. 13b). Further 3D processing yielded a

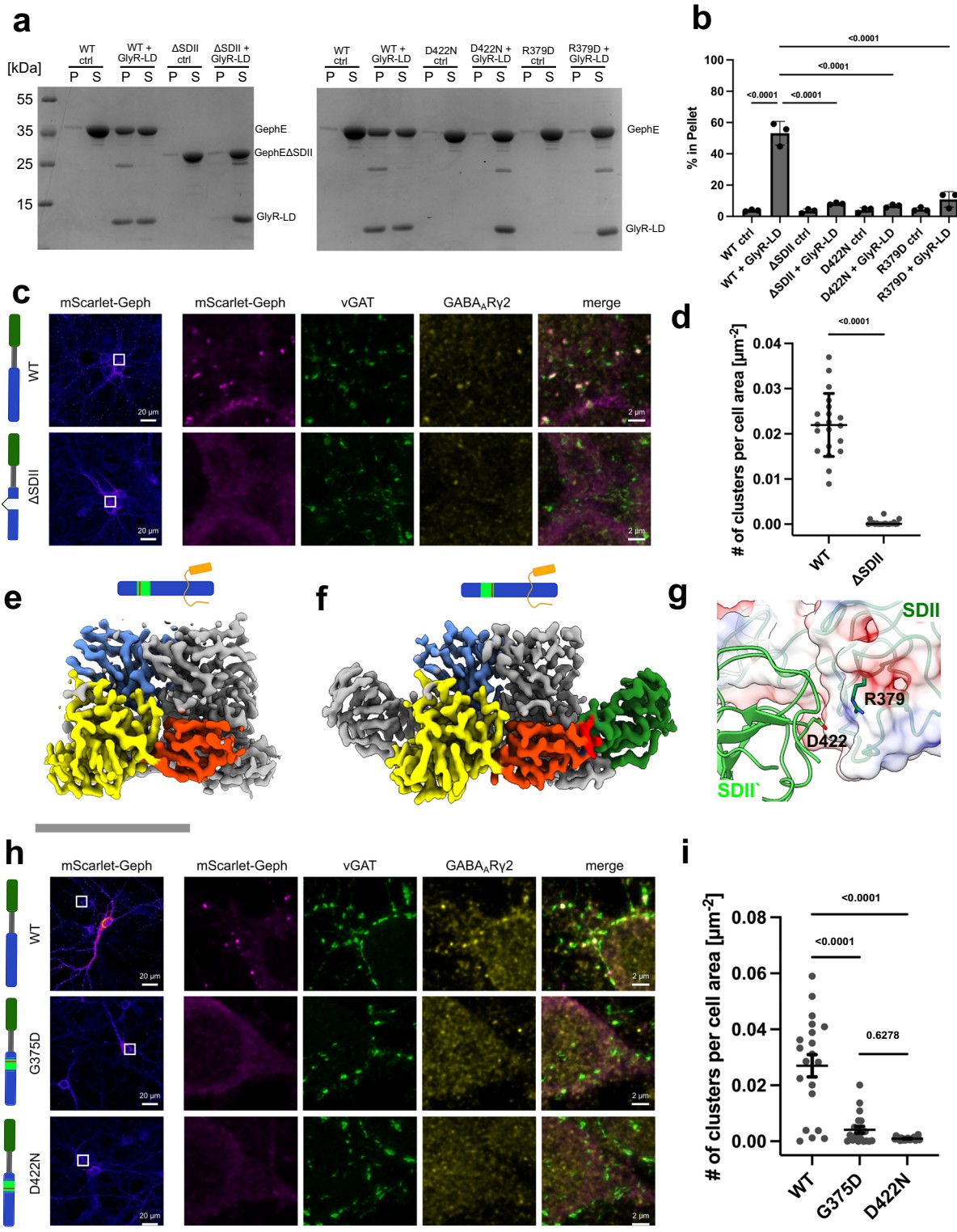

final map of the GephE dimer with a global resolution of 3.2 Å, which, while showing density for the GlyR-loop was lacked density for the SDII region in both subunits (Fig. 5e and Supplementary Fig. 13c, d). Given that neither 3D sorting nor local refinement allowed us to recover density in the SDII region, we hypothesize that the SDII in GephE^G375D is unable to fold properly. This finding is supported by GephE^G375D aSEC experiments showing a larger hydrodynamic radius than GephE^WT

(Supplementary Fig. 1b). In conclusion, GephE^G375D exhibits an unstructured SDII, thus leading to a loss of Geph filament formation.

Next, we expressed and purified the second pathogenic Geph variant, GephE^D422N, and prepared it for cryo-EM. The resulting micrographs were of high quality, showing numerous evenly distributed particles (Supplementary Fig. 14a). Unsupervised 2D classification yielded high-quality class averages showing dimeric structures from

**Fig. 5 | Proper SDII folding and build-up of Gephyrin filaments are essential for synaptic cluster formation and LLPS. a** Representative SDS-PAGE of sedimentation experiment showing the amount of protein present in pellet (P) and supernatant (S) fractions of GephE variants (WT, ΔSDII, D422N, R379D) either without GlyT-LD (ctrl) or with (+GlyR-LD). **b** Quantification of band intensities in the pellet fraction; $n = 3$ presented as means ± SD. Data were analyzed by one-way ANOVA ($F_{9,20} = 47.78$, $p = 1.42 \cdot 10^{-11}$), Tukey's post hoc test; $p$-values are indicated. **c** Representative images of gephyrin$^{floxed/floxed}$ murine hippocampal neurons infected with moxBFP-Cre and either mScarlet-Geph$^{WT}$ or mScarlet-Geph$^{ΔSDII}$ (pseudo-colored fire LUT, magenta) and immunostained against vesicular GABA transporter (vGAT, green) and GABA$_A$R γ2 subunit (yellow). **d** Quantification of gephyrin clusters from c per cell area; $n = 20$ images per condition from four independent cultures. Each data point represents cluster density per acquired image, means ± SD are indicated. Data were analyzed by unpaired, two-tailed Welch's two-sample $t$-test ($t(19.3) = 13.8$, $p = 1.92 \cdot 10^{-11}$). **e** Surface representation of GephE$^{G375D}$ dimer highlighting the positions of SDI (red), SDIII (yellow) and SDIV (blue) of one monomer.

The second monomer is shown in gray. Scale bar 50 Å. **f** Surface representation of GephE$^{D422N}$ dimer highlighting the positions of SDI (red), SDII (green) SDIII (yellow) and SDIV (blue) of one monomer. The second monomer is shown in gray. Scale similar to (**f**). **g** Close up depiction of the SDII interface highlighting D422 situated in SDII' and R379 in close proximity situated in SDII. **h** Representative images of gephyrin$^{floxed/floxed}$ murine hippocampal neurons infected with moxBFP-Cre and either mScarlet-Geph$^{WT}$, mScarlet-Geph$^{G375D}$ or mScarlet-Geph$^{D422N}$ (pseudo-colored fire LUT, magenta) and immunostained against vesicular GABA transporter (vGAT, green) and GABA$_A$R γ2 subunit (yellow). **i** Quantification of gephyrin clusters from h per cell area; $n = 20$ images per condition from four independent cultures. Each data point represents cluster density per acquired image; means ± SD are indicated. Data were analyzed by one-way ANOVA ($F_{2,57} = 34.94$, p $= 1.25 \cdot 10^{-10}$), Tukey's post hoc test; $p$-values are indicated. Cartoon bar diagrams illustrate the Geph constructs used in this experiment (green: GephG, gray: GephC, light green: SDII, red: position of amino acid substitution).

various angles, but no indications of filament formation. Different from GephE$^{G375D}$, the observed structures were similar to those observed for GephE$^{WT}$, including folded SDIIs (Supplementary Fig. 14b, c). Further 3D processing yielded a final map of the GephE dimer with a global resolution of 3.1 Å, which showed density for the SDII region in both subunits, allowing for C2 symmetric 3D reconstruction (Fig. 5f and Supplementary Fig. 14d, e). Importantly, while GephE$^{D422N}$ showed a completely folded SDII, no filament formation was evident either during 2D or 3D processing (Fig. 5f and Supplementary Fig. 14c). Therefore, the substitution of Asp422 by Asn appears to fully abolish SDII-based filament formation. Close inspection of the SDII-SDII' interface in our GephE$^{WT}$ structure identified a possible ion bridge between Asp422 and Arg379 of the neighboring GephE dimer (Fig. 5g). To investigate the potential importance of this hypothesized ion bridge, we created a variant in which Arg379 was substituted with Asp (GephE$^{R379D}$) to reverse the charge and thus disrupt the hypothesized ion bridge. Similar to GephE$^{D422N}$, GephE$^{R379D}$ showed completely folded SDIIs but no filament formation, demonstrating that this ion bridge is a crucial component of the filament interface (Supplementary Fig. 15).

Next, we corroborated that for both the known patient mutant GephE$^{D422N}$ and the mutant GephE$^{R379D}$ the inability to form filaments affects their ability to undergo LLPS in vitro. Indeed, both mutant variants exhibited strongly reduced LLPS formation comparable to Geph$^{ΔSDII.}$ (Fig. 5a, b).

Finally, to corroborate that the inability to form filaments also has physiological relevance, we expressed either mScarlet-tagged Geph$^{G375D}$ or Geph$^{D422N}$ constructs in dissociated murine hippocampal neurons lacking endogenous gephyrin. Similar to mScarlet-Geph$^{ΔSDII}$, mScarlet-Geph$^{G375D}$ and mScarlet-Geph$^{D422N}$ were diffusely localized throughout hippocampal neurons and did not form any postsynaptic clusters (Fig. 5h, i).

Combined, our structural and functional studies of Geph disclosed a hitherto unknown type of oligomerization of GephE dimers, leading to Geph filament formation. This interaction is stabilized by a single GlyR-loop bound to both receptor bindings sites of a GephE dimer. Deleting the filament forming SDII or introducing either known pathogenic mutations or targeted disruption of an ion-bridge, all prohibited GephE filament formation, caused loss in LLPS and impaired synapse formation in neurons.

## Discussion

Our structural and functional data uncovered GephE dimer filament formation as the underlying molecular mechanism for the formation of Geph-containing postsynaptic densities. Negative stain EM data revealed pronounced conformational heterogeneity of full-length gephyrin, consistent with previous studies, which suggest structural variability in purified Geph trimers ranging from so-called 'closed' to

'open' states[20]. The addition of the GlyR-loop not only reduced overall heterogeneity, but also led to the formation of filamentous assemblies of Geph. Cryo-EM data revealed that these filaments are mediated by a hitherto unknown SDII-SDII' interface, which was corroborated by structural data derived both from full-length gephyrin and various GephE variants. Our observation that a single GlyR-loop could bridge two GephE subunits by occupying both previously identified GlyR-loop binding sites simultaneously highlights the important role of receptor binding in GephE dimerization and filament formation: While previous studies have shown that multiple receptor binding sites increase affinity towards gephyrin[24], the here observed binding of a single peptide bridging both receptor sites in GephE would support a receptor-mediated stabilization of GephE dimers, thus probably lowering the concentration threshold required for filament formation. Notably, gephyrin variants with either a deleted SDII or with single residue substitutions in SDII derived either from epileptic patients or designed based on our atomic model were collectively unable to form GephE filaments, to engage in LLPS or to form postsynaptic clusters. Neither deletion of SDII nor the pathogenic amino acid substitutions affected Geph's ability to bind the receptor drastically, with only minor loss in affinity observed[14,33]. This finding highlights that the underlying cause of the pathological loss of gephyrin function is most likely the inability to form Geph filaments. Given that filament formation requires a critical interplay of GephE dimers, the lack of filament formation by Geph$^{G375D}$ variant further explains its dominant nature in the patient with the hemizygous *GPHN* mutation[14].

Previous electron tomography (ET) has uncovered that the PSD of inhibitory synapses forms a thin layer of 5-15 nm width underneath the postsynaptic membrane[51]. Cryo-ET revealed a mesophasic organization of inhibitory receptors in the postsynaptic membrane, with most receptors having four or fewer neighbors assembling into short, curved and presumably branched assemblies[26]. Our structural observation that Geph is able to form flexible filaments, with trimeric GephG forming the basis for GephE dimer-linked filaments and/or branching points, aligns well with the Cryo-ET data. In addition, an underlying filament network at inhibitory PSD was identified in previous ET studies of GABAergic synapses, revealing filaments with a length of approx. 24 to 32 nm[29], which would correlate to filaments consisting of two to three GephE dimers, agreeing with what has been observed in our study. In contrast to the previously proposed hexagonal lattice model, relying on GephG trimer and GephE dimer interactions, gephyrin filaments not only help to understand the observed regular assemblies of inhibitory receptors in the postsynaptic membrane, but also provide further understanding of the mode of action of pathogenic gephyrin variants[14,33]. Our data lend themselves to the conclusion that a hexagonal lattice is neither a requirement nor a precursor for gephyrin filaments, given that gephyrin filaments were also observed for isolated GephE dimers. However, based on studies from human

patients with different forms of neuropsychiatric disorders harboring gephyrin deletions in either GephG, GephC, or GephE, we conclude that proper oligomerization of both GephG and GephE is required for synapse formation[52]. Future studies are needed to structurally resolve the GephG in gephyrin full-length filaments. One possible role is generating branching points that connect gephyrin filaments, thus incorporating elements of the hexagonal lattice model.

Multiple lines of studies have shown specific organization of receptors and PSD protein throughout the synapse, indicating regulatory mechanisms controlling protein density[26,29,30,53]. Receptor ICD binding stabilizes the GephE dimer, thereby fostering filament formation and PSD stabilization. Our data also suggests that in the absence of receptors, Geph retains its ability to self-organize but only at elevated protein concentration. While we did not observe filament formation during SEC experiments, neither in the presence nor in the absence of the GlyR-loop, we detected filaments also in the absence of the GlyR-loop during cryo-EM. In negative stain EM, we only observed filaments in the presence of GlyR-loop. Geph concentration is a main difference between these three experimental approaches, with SEC having the lowest, negative stain-EM an intermediate, and cryo-EM the highest concentration, especially when accounting for a concentration increase due to the blotting process during grid preparation[38]. Considering an ice thickness of approximately 200 Å and a particle density ranging from 300 to 1000 particles per micrograph, as estimated from our data, a Geph concentration between 0.2 mM and 0.7 mM can be assumed[54].

According to high-resolution light microscopy of postsynaptic densities, it was estimated that an inhibitory PSD contains approximately 200–300 gephyrin molecules[12,28]. Based on an average synapse area of 0.07 μm² and a thickness of 0.015 μm, this corresponds to an estimated gephyrin concentration between 0.34 mM and 0.51 mM within the inhibitory PSD, which agrees well with the gephyrin concentration estimated on our cryo-EM grids[51]. Additionally, it can be assumed that the length of gephyrin filaments correlates with increased or decreased local protein concentrations in the PSD[26,53,55]. Local increase in receptor and gephyrin concentration likely influences the formation of nanodomains within the synapse, which are characterized by elevated receptor and PSD protein concentrations, which is an underlying prerequisite for LLPS at the postsynapse[30,53,55].

A structural comparison of apo- and filament-GephE revealed that binding to GlyR-loop alters the position of SDII closer to the position observed in GephE filaments. This observation suggests a GlyR-ICD-induced conformational change rendering SDII more prone to engage in GephE filament formation. This agrees with the previous X-ray structure of GephE, which identified SDII as the most mobile part of GephE[4,8]. As SDII harbors one half of the substrate binding site for gephyrin's enzymatic function, the conversion of molybdopterin-adenylate into molybdenum cofactor[44], this activity may also impact SDII position and thus filament formation. Additionally, a negatively charged and partially disordered region in SDII, opposite to the here identified filament-forming SDII-interface, has been found crucial for LLPS formation, probably impacting the conformational position of SDII within GephE dimers as well[31].

The concept of LLPS as an underlying mechanistic principle in PSD formation has been put forward in recent years[31,32,56,57]. LLPS often requires increased local protein concentration, intrinsically unstructured protein motifs and a defined window of ion strength[31,58–61]. Although the unstructured GephC could not be resolved in our cryo-EM study, we can exclude its direct participation in the protein-protein interaction. It rather enabled the alignment of various GephE dimers derived from different Geph trimers and may play a regulatory role in the prohibition of phase separation at lower concentrations[62,63]. Given that an SDII-deficient Geph variant was unable to form filaments, to engage in LLPS and to form synapses, we conclude that SDII-dependent GephE filaments contribute strongly to the underlying mechanistic principle of Geph LLPS[31]. Furthermore, in line with

previous studies that identified a lack in LLPS for the pathogenic GephE[G375D] variant[31], we could now resolve the underlying cause as a loss in SDII folding, thus prohibiting GephE filament formation while receptor binding remained largely uncompromised. Finally, in the pathogenic variant Geph[D422N] a critical salt bridge within the SDII interaction site has been abolished underlying the functional importance of a precise alignment of both SDII domains during GephE filament formation. In this regard, receptor binding was slightly reduced, but gephyrin's enzymatic function remained unchanged[33].

In summary, our data collectively support the concept of LLPS-driven subcellular compartmentalization and identified GephE filament formation as the underlying molecular principle that assembles gephyrin into the PSD of inhibitory synapses[58,64].

## Methods

### Ethics statement
We complied with all relevant ethical regulations for animal testing and research. Experiments were approved by the local research ethics committees (Germany, Landesamt für Natur, Umwelt und Verbraucherschutz Nordrhein-Westfalen, reference 2021.A450).

### Expression constructs
6His-tagged Geph, GephE WT and GlyR β-loop-intein expression vectors have been described before[14,23]. 6His-tagged GephE[G375D] and GephE[D422N] expression vectors have been described before[14,33]. For the 6His-tagged GephΔSDII (residues: 1-383, GSGSG-linker, 473-751) variant, the previously described 6His-tagged Geph construct was used as template[23]. Deletion of SDII (residue: 383-473) and insertion of GSGSG-linker was achieved by Gibson Assembly using DpnI restriction enzyme and the following primers (fw-GGCTCAGGTTCCGGACCCATCGGCCACGACATTAA; rv-CAACCTACCCCCATTCCCAGGCTCAGGTTCCGGA). For the 6His-tagged GephE[R379D] variant, the previously described 6His-tagged GephE WT construct was used as template[23]. Side-directed mutagenesis for GephE[R379D] was achieved by PCR with the following primers (fw-CTATGCTGTTGATGCTGCTGATGGT; rv-ACCATCAGCAGCATCAACAGCATAG) followed by DpnI restriction enzyme digest. GlyR-DL and GABAAR-α1/3-DL were kindly provided by the Zhang lab[31].

### Protein expression and purification
All gephyrin variants used in this study were recombinantly expressed in *E. coli* BL21 (DE3) Rosetta cells in 1x LB medium containing 0.1 mg/ml ampicillin. Prior to expression, cells were grown until an OD600 of 0.3–0.4 was reached. His-tagged gephyrin variants were expressed for 16 h at 18 °C. His-tagged GephE WT was expressed for 5 h at 25 °C, whereas GephE variants were expressed for 16 h at 18 °C. Protein expression was induced by the addition of isopropyl-β-D-thiogalactopyranosid (IPTG) to an end-concentration of 250 μM in the cell culture. After protein expression, cells were harvested by centrifugation at 4000 × *g* and resuspended in lysis buffer (50 mM Tris/HCl, pH 7.5, 250 mM NaCl, 5 mM β-mercaptoethanol, 1x protease inhibitor (cOmplete, Roche, Basel, Switzerland) and lysozyme from chicken egg (Sigma-Aldrich) and stored at −20 °C for further use. For purification, the cells were lysed mechanically using EmulsiFlex (Avestin, Ottawa, Canada) and ultrasound for 3 × 30 s at 45% amplitude. Cytosolic extract was separated from cell debris by centrifugation at 50,000 × *g* for 45 min. Prior to Ni-NTA chromatography, the cell extracts were supplemented with imidazole to 20 mM final concentration. The cell extract was applied to 10 ml Ni-NTA resin, and proteins were washed with 10 column volumes (CV) of lysis buffer and 10 CV of lysis buffer containing 40 mM imidazole. Proteins were eluted with 3 CV lysis buffer containing 300 mM imidazole. All proteins were further purified via size exclusion chromatography (SEC) (Superdex 16/600 prep grade, GE Healthcare) using protein buffer (25 mM Tris/HCl, pH 7.5, 250 mM NaCl, 5 mM β-mercaptoethanol, 5% glycerol). All purification steps were performed at 4 °C. GlyR-βloop was expressed

and purified as stated previously[23]. GlyR-βLD, GABA$_A$R-αLD and GABA$_A$R-α3LD were purified according to the protocol provided by Bai and colleagues[31].

## Isothermal titration calorimetry (ITC)

All experiments were performed with a MicroCal Auto-ITC200 (Malvern, Malvern, United Kingdom) in ITC buffer (25 mM Tris pH 7.5, 250 mM NaCl, 5 mM β-mercaptoethanol, 5% glycerol. The sample cell was filled with gephyrin or GephE variants at a concentration of 20–30 μM. GlyR-βloop was filled into the syringe at a concentration of 250–350 μM. Experiments were conducted at 37 °C with an injection volume of 1.5–2 μl, a set reference power of 5 μcal/s, spacing time of 180–300 s and a stirring speed of 750 rpm. Data was analyzed using Origin 7 (OriginLab Corporation), and binding parameters were derived by applying a one-site or two-site binding model.

## Sedimentation assay of in vitro phase separation

For sedimentation assays, proteins were mixed at designated combinations and conditions in a 100 μl final volume in assay buffer (25 mM Tris, pH 7.5, 150 mM NaCl). After 10 min incubation at room temperature, the mixtures were centrifuged at 16,000 × g for 10 min at 25 °C. Right after centrifugation, the supernatant and pellet were separated. The pellet was brought to the same volume as the supernatant. Proteins recovered in supernatant and pellet were analyzed by the Bradford assays or SDS-PAGE with Coomassie Blue staining. The band intensities in SDS-PAGE gels were quantified by ImageJ software. Relative and absolute amounts of proteins in supernatant and pellet were calculated based on the input amounts and relative band intensities. Data from three repeats were presented as means ± SD.

## Negative stain sample preparation and data collection

For negative stain EM, all samples were SEC purified prior to grid preparation using either a Superdex 6 increase 10/300 GL or Superose 6 increase 10/300 GL column with grid buffer (25 mM Tris pH 7.5, 250 mM NaCl, 25 mM Arg/Glu, 5 mM β-mercaptoethanol, 1.5% sucrose). For grid preparation, protein samples were diluted to a final concentration of 0.005 mg/ml using grid buffer. Continuous carbon grids (Quantifoil, Cu 200 mesh) were subjected to glow discharge using a Zepto Plasma Cleaner (Diener) for 30 s to clean the surface and render it hydrophilic. 3 μl protein solution was incubated on the grid for approximately 1 min at room temperature, and then stained with uranyl formate as described elsewhere[65]. The grids were stored in the dark in a dry atmosphere until further use. EM data were acquired using a Talos L120C (Thermo Fisher Scientific) electron microscope equipped with a LaB$_6$ emitter operated at 120 kV. Images were collected automatically using *EPU* (version 2.12.1.2782REL, Thermo Fisher Scientific) on Ceta16M CMOS detector with a calibrated pixel size of 1.86 Å per pixel. Defocus values were set to range from −0.3 to −2.0 μm.

## Cryo-EM sample preparation and data collection

Prior to grid preparation, all samples were isolated using a Superdex 6 increase 10/300 GL column with grid buffer. Afterwards, samples were diluted to a concentration of 0.5 mg/ml using the grid buffer. Protein samples were applied to a glow-discharged UltrAUfoil R1.2/1.3 Au300 (Quantifoil Micro Tools GmbH), incubated for 5 s, blotted for 3.5 s and flash-frozen in liquid ethane using a Vitrobot Mark IV device (Thermo Fisher Scientific) set to 100% humidity at 21 °C. Grids were stored under liquid nitrogen conditions until usage. Cryo-EM data were acquired using a Titan Krios G3i (Thermo Fisher Scientific) electron microscope operated at 300 kV with either 0° or 25 ° stage tilt. Images were collected automatically using EPU (version 2.12.1.2782REL, Thermo Fisher Scientific) on a Falcon III direct electron detector with a calibrated pixel size of 0.862 Å px$^{-1}$. Movies were collected in counting mode with a total dose of approximately 30–40 e$^-$/Å$^2$ distributed among 42 frames. Defocus values ranged from −0.5 to −2.6 μm.

## EM image processing

Image processing was performed using cryoSPARC (version 4.4)[66]. Movie stacks were first corrected for drift and beam-induction motion, and then used to determine defocus and other CTF-related values. Only high-quality micrographs with low drift metrics, low astigmatism and good agreement between experimental and calculated CTFs were further processed. On these high-quality micrographs, putative particles were automatically picked based on expected protein diameter, extracted, and subjected to reference-free 2D classification. Representative 2D classes were then used for a template-based picking approach, particles extracted again, subjected to reference-free 2D classification to exclude artefacts, and subsequent 3D classification using C1 symmetry to identify high-quality particles. 3D classification was performed with a target resolution of 6 Å in order to ensure that secondary-structure elements are also included as a feature in the classification algorithm. Particles showing protein-like density features, especially features resembling secondary structure elements, were further refined using the non-uniform refinement strategy. Subsequent data processing workflows differ from this point onwards between the different datasets (For more information, see supplementary information).

## Atomic model building

The atomic model was built starting from the gephyrin E domain dimer crystal structure (PDB ID 2FTS) and the gephyrin amino acid sequence (UniProt Q9NQX3). First, Coot (version 0.9.8.02)[67] was used to semi-manually fit amino acids into a single subunit of the EM density map. Next, Phenix (version 1.21)[68] was used to generate a non-crystallographic symmetry (NCS)-related second subunit and both subunits were then simultaneously refined against the 3D density map. This process was iterated until the fit to the density map and geometric parameters converged. The final atomic model accounts for residues 31 to 556. For further details and statistics, see Supplementary Table 1. Molecular visualization and analysis were done using UCSF ChimeraX (version 1.9)[69]

In the final atomic model, the following amino acid sequences were reduced to their respective C-alpha: ChainA/LEU 432-ILE 447, ChainA/VAL 574-ASP 580, ChainB/VAL 574-ASP 580, ChainC/LEU 432-ILE 447.

## DNA constructs

mScarlet-Gphn C4c from pAAV-hSyn-fDIO-mScarlet-Gphn_C4c (Addgene plasmid # 194974; RRID:Addgene_194974) and pAAV-hSyn-fDIO-mScarlet-Gphn_P1 (Addgene plasmid # [194972]; RRID: Addgene_194972) were initially subcloned into pGP-AAV-syn-FLEX-jGCaMP7s, in which the WPRE was removed using HindIII and NotI (pGP-AAV-syn-FLEX-jGCaMP7s-WPRE was a gift from Douglas Kim & GENIE Project (Addgene plasmid # 104491; RRID:Addgene_104491)) using NEBuilder HiFi DNA Assembly (NEB). Since one of the AAV2 ITR sites was destroyed in the process, the inserts mScarlet-Gphn C4c and mScarlet-Gphn_P1, including loxP and lox2272 sites, were subcloned into the backbone of pAAV-hSyn-fDIO-mScarlet-Gphn_P1 (Addgene plasmid # 194972; RRID:Addgene_194972) using HindIII and Eco47III. His-tagged Gphn C4c (P2) ΔSDII in pQE80 was generated by site-directed mutagenesis using forward GGCTCA GGTTCCGGACCCATCGGCCACGACATTAA and reverse TCCGGAA CCTGAGCCTGGGAATGGGGGTAGGTTG primers. His-tagged Gphn (P1) D422N in pQE80 was generated by site-directed mutagenesis using forward TGCGGTGCTAATGCAGTAGTG and reverse CACTACTGCAT-TAGCACCGCA primers. His-tagged Gphn (P1) G375D in pQE80 was generated by site-directed mutagenesis using forward CATCAG-TAAAAGATGACTATGCTGTTCG and reverse CGAACAGCATAGT CATCTTTTACTGATG primers. Gphn C4c (P2) ΔSDII, Gphn (P1) D422N, and Gphn (P1) G375D were subcloned into the backbone of pAAV-hSyn-FLEX-mScarlet-Gphn C4c or pAAV-hSyn-FLEX-mScarlet-Gphn P1 using

XhoI and BshTI restriction sites. pAAV-CamKIIa-moxBFP-P2A-Cre was generated in two steps. First, pAAV-CamKIIa-moxBFP-Cre was generated by subcloning Cre from pAAV.CMV.HI.eGFP-Cre.WPRE.SV40 (a gift from James M. Wilson (Addgene plasmid # 105545; RRID:Addgene_105545)) into pAAV-CamKIIa-moxBFP-WPRE (Addgene plasmid # 194976; RRID:Addgene_194976). In the second step, the P2A site was inserted in between moxBFP and Cre via site-directed mutagenesis using forward GGCGACGTGGAGGAGAACCCCGGCCCGCCGG CAGCATGTCCGGAGAGCAAAAGCTG and reverse primers TCTCCTCC ACGTCGCCGGCCTGCTTCAGCAGGCTGAAGTTGGTGGCGCCGCTGCC CTTGTACAGCTCGTCCATGC. The plasmids pAdDeltaF6 (Addgene plasmid # 112867; RRID:Addgene_112867) and pAAV2/1 (Addgene plasmid # 112862; RRID:Addgene_112862) were gifts from James M. Wilson. All DNA constructs were confirmed by Sanger sequencing (Eurofins), and the integrity of AAV2 ITR sites was confirmed by analytical SmaI restriction digest.

### Primary hippocampal cultures

Floxed gephyrin mice[48] were a gift from Ralph A. Nawrotzki. Floxed gephyrin mice were back-crossed to C57BL/6NRj. Mice were housed at $22 \pm 2\,°C$, with a humidity of $55 \pm 10\%$, on a 12 h light-dark cycle with free access to food and water.

Dissociated primary hippocampal cultures were prepared from homozygous gephyrin floxed embryos (E17.5) of either sex, which were obtained from a total of four homozygous parental pairs. Per preparation, all embryonal hippocampi obtained from one pregnant mouse were pooled prior to cell dissociation. After dissociation, 65,000 cells were seeded on Poly-L-lysine-coated 13-mm cover slips. Cultures were grown in 0.5 ml per well neurobasal medium supplemented with B-27, N-2, GlutaMAX™ (Thermo Fisher Scientific), and Pen/Strep. Recombinant AAV1 particles were diluted in Neurobasal medium with supplements before they were added to the cultures. Cultures were infected after 4 days in vitro (DIV) with $1 \times 10^8$ viral genome copies (GC) of moxBFP or moxBFP-P2A-Cre and after 8 DIV with $5 \times 10^8$ GC of either AAV-hSyn-FLEX-mScarlet-Gphn C4c. On DIV 11, 50% media was exchanged for fresh neurobasal media with supplements.

### rAAV2/1 preparation

Recombinant AAV2/1 particles were prepared in HEK293T cells (DSMZ no. ACC 635) according to the protocol by Kimura and colleagues[70]. Viral particles were precipitated with PEG/NaCl and subsequently cleared via chloroform extraction[70]. Purity of viral preparations was assessed with SDS-PAGE, and AAV titers were determined using Gel green® (Biotium) protocol[71]. Fluorescence was measured at $507 \pm 5$ nm excitation and $528 \pm 5$ nm emission using a plate reader equipped with monochromators (Tecan Spark).

### Immunocytochemistry

Cells were fixed at DIV14 with 4% formaldehyde in PBS for 15 min and quenched with $NH_4Cl$ in PBS at room temperature (RT). All subsequent incubation and wash steps (PBS) were performed at RT. After blocking/permeabilization for 1 h with 10% goat serum, 1% BSA, 0.2% Triton X-100 in PBS, the following primary antibodies (Synaptic Systems) were used: anti-vesicular GABA transporter (vGAT) (1:1000, #131003) for inhibitory presynaptic terminals; anti-GABA$_A$R γ2 (1:500, #224004) for postsynaptic GABA$_A$Rs. The following secondary antibodies were used: goat anti-rabbit AlexaFluor 488 (1:500, #A-11034, Thermo Fisher Scientific) and goat anti-guinea pig AlexaFluor 647 (1:500, #ab150187, Abcam). Coverslips were mounted with Mowiol/Dabco.

### Confocal microscopy and image analysis

Image stacks [0.3-μm z-step size, $3432 \times 3432$ ($144.77 \times 144.77$ μm)] were acquired on a Leica TCS SP8 LIGHTNING upright confocal microscope with an HC PL APO CS2 63×/1.30 glycerol objective,

equipped with hybrid detectors (Leica HyD) and the following diode lasers: 405, 488, 552, and 638 nm. LIGHTNING adaptive deconvolution using "Mowiol" setting was applied. Images were segmented and analyzed in an automated fashion using ImageJ/FIJI 1.53c (https://imagej.net/software/fiji/), and the code is available online at GitHub (https://github.com/FilLieb/quantitative_synapse_analysis).

### Statistics and reproducibility

Micrographs shown for the cryo-EM experiments are representative for the respective dataset. Statistical analyses were performed using R version 4.2.2 (2022-10-31 ucrt), readxl v. 1.4.1, rstatix v. 0.7.1, tidyverse v. 1.3.2 in Rstudio environment (https://www.rstudio.com/), Platform: x86 64-w64-mingw32/x64 (64-bit). Exact $p$-values can be found in the source data.

### Reporting summary

Further information on research design is available in the Nature Portfolio Reporting Summary linked to this article.

## Data availability

Unless otherwise stated, all data supporting the results of this study can be found in the article, supplementary, and source data files. Plasmids have been made available via Addgene [https://www.addgene.org/]. The EM maps for GephE$^{WT}$, GephE$^{G375D}$, GephE$^{D422N}$, and GephE$^{R379D}$ have been deposited in the EMDB under accession codes EMD-51644, EMD-54820, EMD-54821, EMD-54824. The raw electron microscopy imaging data are available on request due to their large size. Atomic coordinates for GephE$^{WT}$ have been deposited in the Protein Data Bank under the accession code 9GW9. Source data are provided with this paper.

## Code availability

The code used for the image analysis is available at GitHub [https://github.com/FilLieb/quantitative_synapse_analysis].

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

## Acknowledgements
We thank the Biocenter Imaging Facility for their support with confocal microscopy, Dr. Franziska Neuser and Julia Reich, for their support with primary hippocampal cultures, as well as Monika Laurien, Simona Jansen and Jennifer Lange for their technical support. We thank PD Dr. med. Ralph A. Nawrotzki, Universität Heidelberg, for kindly sharing floxed gephyrin mice with us. We thank the StruBiTEM facility for access to the cryo-EM infrastructure (Cologne, funded by DFG Grant INST 216/949-1 FUGG) and the RRZK for access to the computing infrastructure of CHEOPS (Cologne, funded by DFG Grant INST 216/512/1 FUGG). Funding by the German Research Foundation is gratefully acknowledged by GS (RTG2550/1736 project ID 411422114). Funding by the Ministry of Culture and Science of the State of North Rhine-Westphalia is gratefully acknowledged by EB (iHEAD MKW NRW, PB22-025A). We also thank Prof. Mingjie Zhang, Hong Kong University of Science and Technology, for kindly sharing their DNA constructs with us.

## Author contributions
A.M., E.B., and G.S. designed the research study; A.M., F.L., E.H.W.B., I.v.S., and K.B. conducted experiments; A.M., F.L., E.H.W.B., N.B., K.B., and M.G. acquired data; A.M., F.L., E.H.W.B., N.B., M.G. analyzed data; A.M. performed structural model building; A.M., F.L., E.B., and G.S. wrote the manuscript with input from all coauthors.

## Funding

## Competing interests
The authors declare no competing interests.
