## [Transparent Peer Review file · Nature Communications]

Gephyrin filaments represent the molecular basis of inhibitory postsynaptic densities

Corresponding Author: Professor Guenter Schwarz

Version 0:

Reviewer comments:

Reviewer #1

(Remarks to the Author)

This study, contributed by the lab with extensive past structural and biochemical research on gephyrin, elucidated the structural basis of gephyrin-mediated inhibitory postsynaptic density (PSD) formation through cryo-EM and functional analyses. The study revealed that gephyrin E-domain dimers polymerize into flexible filaments via a Z-shaped SDII-SDII' interface stabilized by GlyR β -loop binding. The authors demonstrated that a single GlyR β -loop bridges both binding pockets within a GephE dimer, and the pathogenic mutations (G375D, D422N) disrupt filament assembly through distinct mechanisms - by SDII misfolding (G375D) or by breaking a critical SDII-SDII' salt bridge (D422N) - despite retaining the receptor binding capability. These filaments were shown to be essential for liquid-liquid phase separation (LLPS) in vitro and synaptic clustering in neurons, providing a molecular explanation for how gephyrin organizes inhibitory synapses and how its disruption leads to epileptic encephalopathy. The work advances understanding of inhibitory synapse organization by providing a novel dynamic gephyrin filament-based mechanism and also raising important new questions about regulation and disease mechanisms related to gephyrin for future studies. As such, this is a strong candidate for NC with minor revisions.

Below are detailed comments and suggestions for improvement:

Major Comments

1. Physiological Relevance of Filament Formation

The authors noted that filament formation occurs at high protein concentrations during cryo-EM grid preparation. While they argue synaptic concentrations of gephyrin (~0.34-0.51 mM) align with these conditions, further discussion is needed to address whether this assembly occurs in vivo or represents an in vitro artifact. Could other synaptic components (e.g., tubulin, collybistin) modulate filament stability? In vivo validation (e.g., super-resolution imaging of SDII-dependent filament organization in neurons) would greatly strengthen the conclusion.

2. Resolution and Model Accuracy

The SDII-SDII' interface is resolved at 3.5-4.0 Å, with fragmented density in chain B. Deep-learning density sharpening was applied, the model's reliability in this region could benefit from cross-validation. Mutagenesis targeting the proposed D422-R379 salt bridge (e.g., disrupting/reversing charges) followed by biochemical or structural assays (filament formation/LLPS in vitro) will confirm the interface's functional importance. The authors already included data for the D422N mutation, additional mutagenesis on the counterpart (e.g., R379A/D) can further provide compelling validation.

3. Gephyrin Trimer Stability vs. Dissociation:

The authors describe gephyrin as a trimer mediated by its G-domain (GephG), consistent with prior biochemical and structural studies. However, the cryo-EM and negative stain EM data reveal a striking discrepancy: most particles showed isolated E-domain (GephE) dimers, only a minor subset (~17%/35% in negative stain EM) corresponds to the expected GephG trimer alongside GephE dimer (Fig. 1c/e). Could the trimeric state be destabilized during the purification or grid preparation?

The authors attributed the gephyrin filament formation to the GephE dimer, but how does the trimeric GephG scaffold contribute? If GephG trimers are sparse in the data, does this imply filaments are primarily E-domain-driven, with G-domains playing a secondary role? Would GephE alone also form filaments?

The blotting-induced concentration during cryo-EM grid preparation might favor GephE dimerization over trimer maintenance. A control experiment with crosslinking prior to grid preparation could test the role of GephG in filament formation.

4. Comparison to Hexagonal Lattice Model

The study dismisses the hexagonal lattice model (Saiyed et al., 2007) but does not directly contrast it with the filament model. A discussion of how these two models may (e.g., filaments as a dynamic precursor to lattices) or may not be mutually compatible would strengthen the paper.

Minor Comments

5. Receptor Binding in the G375D Mutation

The authors reported that GephE-G375D was purified and analyzed by cryo-EM in the presence of a 5-fold molar excess of GlyR β -loop. However, Figure 5e (G375D dimer structure) shows no visible density for the GlyR β -loop, unlike the WT GephE-GlyR β -loop complex (Fig. 4). The absence of the receptor density in the mutant is puzzling and requires clarification.

6. Specified the Molecular Weight of the Geph-GlyR Complex

The manuscript states that the Geph-GlyR β -loop complex elutes in SEC at the expected molecular weight of a trimeric complex (Sup-Fig 1a). However, a more rigorous molecular weight determination would strengthen this claim, as SEC elution volume alone is not definitive due to shape and dynamic equilibrium of the complex.

7. Statistical Reporting

Figure 5: Please specify the statistic test used (e.g., unpaired t-test, ANOVA?).

8. Typo

Original problematic sentence: "Next, we asked whether the GephE filaments formation are to LLPS at inhibitory synapses, which has been hypothesised to represent the molecular basis of Geph clustering." A verb is missing between "are" and "to".

Reviewer #2

(Remarks to the Author)

In this manuscript, Macha and colleagues report the structures of Gephyrin E filaments using single-particle cryo-EM analysis. They examine the structures of Geph WT and Geph WT+GlyR at low concentrations (Fig. 1), Geph WT+GlyR at high concentrations where filaments were observed (Fig. 2a-b), GephE+GlyR filaments (Fig. 2c-e, 3-4), full-length Geph Δ SDII-GlyR, and GephE G375D/GephE D422N (Fig. 5). The structural study is well-executed and comprehensive. The observation of GephE filament formation and the interlocking interface of neighboring SDIIs is intriguing. Furthermore, the finding that SDII deletion abolishes filament formation, phase separation, and synaptic receptor clustering provides valuable insights into Gephyrin function. The analysis of disease-linked mutants further strengthens the study's significance.

However, the physiological relevance of GephE filaments remains unclear. Additionally, the connection between filament formation, phase separation, and synaptic receptor clustering appears somewhat premature, as further evidence is needed to confirm Gephyrin's ability to undergo phase separation both in vitro and in neurons. Below are specific points for consideration:

1. As the authors note, GephE filaments form during cryo-EM sample preparation and can even assemble in the absence of GlyR. This raises questions regarding their physiological relevance and whether such filaments can form in a cellular environment.
2. The formation of filaments or LLPS typically requires multivalent interactions. For GephE, in the presence of GlyR, these interactions may be mediated by GlyR-induced dimerization and SDII interactions. In full-length Geph, additional multivalent factors, such as GephG trimerization, could facilitate assembly even in the absence of GlyR. Thus, the filament of the full-length Geph and Geph E might have different structures. Given that the authors successfully addressed preferred orientation in GephE filament analysis, could the same strategy be applied to solve the filament structure of full-length Geph, which may be more physiologically relevant?
3. In Figure 1e-g, is it possible to resolve GlyR in these structures? How does its structure compare to that of GlyR in the filament?
4. On page 9, last line, should "GephE" be "GlyR"?
5. The only evidence provided for Geph LLPS is a sedimentation assay. Filament formation is distinct from LLPS, which, by definition, involves liquid-liquid phase separation. Cryo-EM structural data on filaments do not constitute evidence for LLPS. To support LLPS, the liquid properties of the assembly should be demonstrated, for example, through imaging (e.g., round morphology), fusion events, or FRAP analysis.
6. Additional evidence could also strengthen the claim of Geph undergoing LLPS in cells. For instance, a FRAP experiment could assess the dynamics of Geph clusters in Figure 5A and H.
7. In Figure 5A and H, Geph foci colocalize with vGAT foci, which persist even in the presence of Geph Δ SDII, despite its inability to form foci. This raises the question of whether full-length Geph spontaneously forms foci via LLPS or is merely recruited into vGAT foci through vGAT binding.
8. Could the reduced foci formation observed for GephE G375D/GephE D422N be due to reduced vGAT binding rather than impaired assembly? Additionally, it would be helpful to confirm whether these mutants retain their ability to bind the GlyR

loop, though this has been previously reported (references 31 and 33).

9. The cryo-EM study on disease-linked mutants was conducted using the GephE domain alone. To better understand and compare their assembly properties, whether through filament formation or LLPS, it would be beneficial to perform sedimentation assays and/or cryo-EM analysis using full-length Geph with these mutations.

10. Currently, there is insufficient evidence to support the claim that Geph forms filaments or fiber-like structures in cells. Some statements should be revised accordingly, or additional experimental data should be provided. For example, on page 12, lines 2-3, the manuscript states: "Overall, these experiments demonstrate that the SDII domain-mediated filaments are a prerequisite for the formation of postsynaptic clusters at GABAergic synapses." This conclusion would be more compelling with stronger supporting evidence.

11. The statement, "Next, we asked whether the GephE filaments formation are to LLPS at inhibitory synapses..." is unclear and should be revised for clarity.

12. In the Methods section, there is a heading for "Sedimentation and imaging assays of in vitro phase separation," but no imaging assays are included. If imaging data were not performed, the heading should be revised accordingly.

13. In the Methods section, there is a typo in "Protein expression and purification" (should be "purification").

14. In the Discussion, the manuscript states that for D422N, "receptor binding and Gephyrin's enzymatic function remained largely unchanged," citing reference 33. However, reference 33 (Macha et al.) appears to show that Geph D422N has reduced GlyR binding. This discrepancy should be clarified.

Version 1:

Reviewer comments:

Reviewer #1

(Remarks to the Author)

The authors have performed a very nice round of revision. The revised manuscript has addressed all of the comments raised by this reviewer. Accordingly, this reviewer supports publication of the manuscript in NC.

Reviewer #2

(Remarks to the Author)

The authors have satisfactorily addressed all of my previous comments and concerns in the revised manuscript. The additional clarifications and revisions have strengthened the work, and I have no further concerns. I support the publication of this paper in its current form.

REVIEWER COMMENTS

Reviewer #1 (Remarks to the Author):

This study, contributed by the lab with extensive past structural and biochemical research on gephyrin, elucidated the structural basis of gephyrin-mediated inhibitory postsynaptic density (PSD) formation through cryo-EM and functional analyses. The study revealed that gephyrin E-domain dimers polymerize into flexible filaments via a Z-shaped SDII-SDII' interface stabilized by GlyR β -loop binding. The authors demonstrated that a single GlyR β -loop bridges both binding pockets within a GephE dimer, and the pathogenic mutations (G375D, D422N) disrupt filament assembly through distinct mechanisms - by SDII misfolding (G375D) or by breaking a critical SDII-SDII' salt bridge (D422N) - despite retaining the receptor binding capability. These filaments were shown to be essential for liquid-liquid phase separation (LLPS) in vitro and synaptic clustering in neurons, providing a molecular explanation for how gephyrin organizes inhibitory synapses and how its disruption leads to epileptic encephalopathy. The work advances understanding of inhibitory synapse organization by providing a novel dynamic gephyrin filament-based mechanism and also raising important new questions about regulation and disease mechanisms related to gephyrin for future studies. As such, this is a strong candidate for NC with minor revisions.

We thank the reviewer for the kind interpretation of our work and suggesting it as a strong candidate for publication. We hope that we have successfully addressed their concerns in our revised manuscript. Please find below our point-by-point response to the reviewers in blue font.

Below are detailed comments and suggestions for improvement:

Major Comments

1. Physiological Relevance of Filament Formation

The authors noted that filament formation occurs at high protein concentrations during cryo-EM grid preparation. While they argue synaptic concentrations of gephyrin (~0.34-0.51 mM) align with these conditions, further discussion is needed to address whether this assembly occurs in vivo or represents an in vitro artifact. Could other synaptic components (e.g., tubulin, collybistin) modulate filament stability?

ANSWER: With both reviewers (see also comment 1 of reviewer #2) commenting on the physiological relevance of filament formation, we noted that we failed to properly convey that the two gephyrin variants observed in patients with epileptic encephalopathy (G375D and D422N) are fully consistent with a lack of filament formation, which impairs inhibitory synapse formation. Regarding in vivo relevance of filamentous structures, we would like to refer to a work by Linsalata et al. J Comp Neurol (2014) showing electron tomography on γ -aminobutyric acid-ergic synapses that revealed a discontinuous postsynaptic network of filaments.

Regarding other synaptic components, we agree with the reviewer and are happy to mention that we have studied the impact of collybistin, a ligand of gephyrin and brain-specific nucleotide exchange factor, on gephyrin filament formation. This study is part of a separate submission that is currently undergoing major revisions at Nature Communications (<https://www.researchsquare.com/article/rs-5890130/v1>). This submission shows that collybistin binds to different types of sites on the gephyrin filament, thereby controlling the dynamics and size of filament formation.

2. Resolution and Model Accuracy

The SDII-SDII' interface is resolved at 3.5-4.0 Å, with fragmented density in chain B. Deep-

learning density sharpening was applied, the model's reliability in this region could benefit from cross-validation. Mutagenesis targeting the proposed D422-R379 salt bridge (e.g., disrupting/reversing charges) followed by biochemical or structural assays (filament formation/LLPS in vitro) will confirm the interface's functional importance. The authors already included data for the D422N mutation, additional mutagenesis on the counterpart (e.g., R379A/D) can further provide compelling validation.

ANSWER: We agree that the limited resolution due to the inherent flexibility of SDII calls for validation of the proposed salt bridge between D422 and R379. As suggested by the reviewer, we generated the 'complementary' R379D variant and performed functional and structural studies. In the revised manuscript we now present both cryo-EM (Sup-Fig 15) as well as in vitro LLPS formation data (Fig 5 c, d) on the R379D variant that indeed strengthen our hypothesis of the aforementioned salt bridge. The Geph-E R379D variant failed to sediment, rendering it unable to mediate LLPS. In cryo-EM imaging, we only observed Geph-E R379D dimers, with no detectable filaments, under comparable experimental conditions.

3. Gephyrin Trimer Stability vs. Dissociation:

The authors describe gephyrin as a trimer mediated by its G-domain (GephG), consistent with prior biochemical and structural studies. However, the cryo-EM and negative stain EM data reveal a striking discrepancy: most particles showed isolated E-domain (GephE) dimers, only a minor subset (~17%/35% in negative stain EM) corresponds to the expected GephG trimer alongside GephE dimer (Fig. 1c/e). Could the trimeric state be destabilized during the purification or grid preparation?

ANSWER: This comment made us realize that our figures fail to explicitly state which construct resulted in which data. To account for this, we now added a cartoon to each figure showing explicitly whether full-length gephyrin (GephG, C-linker, and GephE) or isolated GephE were used to generate the data (Fig. 1c,e; 2a,c; 5c,e,f,h).

Regarding the questions raised by reviewer #1, we attribute the smaller number of particles displaying GephE dimer density together with GephG trimer density to the inherent flexibility of the C-linker that connects the two domains. Consequently, only a subset of particles will have GephE and GephG in the same relative orientation, resulting in 2D classes with clear density for both domains. Particles end up in 2D classes with defined GephE dimers and in classes not centered on GephG trimers because the GephG trimer is smaller than the GephE dimer. This makes it easier for the alignment/classification algorithms to focus on the GephE dimer. Therefore, we do not believe that the trimeric state is destabilized during purification or grid preparation. However, to validate this assumption, we performed negative stain EM of cross-linked samples of full length gephyrin validating that indeed GephG is not affected by our sample preparation workflow (see Figure R1 in this letter).

Figure 1: Glutaraldehyde crosslinking of full-length gephyrin trimer reveals stable GephG trimer

(a) Chromatogram of crosslinked and non-crosslinked GephWT trimers. Crosslinked sample was used for grid preparation. SDS-PAGE was used to confirm crosslinking. (b) 2D-class averages of crosslinked GephWT trimer negative stain dataset. Classes show a clearly defined GephG trimer at different positions relative to the GephE dimer. Only two GephE domain are resolved in these classes. (c,d) Refined 3D-classes of crosslinked GephWT trimer negative stain dataset. Volumes were low-pass filtered to 18 Å. GephE dimer (PDB: 2FU3, blue) and GephG trimer (PDB: 2JLJ, red) were fitted into both classes.

The authors attributed the gephyrin filament formation to the GephE dimer, but how does the trimeric GephG scaffold contribute? If GephG trimers are sparse in the data, does this imply filaments are primarily E-domain-driven, with G-domains playing a secondary role? Would GephE alone also form filaments?

The blotting-induced concentration during cryo-EM grid preparation might favor GephE dimerization over trimer maintenance. A control experiment with crosslinking prior to grid preparation could test the role of GephG in filament formation.

ANSWER: We thank the reviewer for highlighting that one of the main points of our manuscript was not shown clearly enough: indeed, we interpreted our data in a way that filaments are E-domain-driven with the G-domain being dispensable for filament formation. Our reason for this assumption is that the sample containing isolated GephE forms filaments in the same way as full-length gephyrin does (Figure 2). In addition to the aforementioned cartoon figures aiming to explicitly shown the construct/gephyrin variant used for each experiment, we have amended our text to make this clearer for the reader in each section.

Moreover, in the revised manuscript we now also present non-averaged data (Sup-Fig 6b1) directly showing the filaments formed by GephE in the absence of GephG. We hypothesize that dimer formation of GephE is essential for and sufficient to form filaments which are formed more readily at higher concentrations. Binding to glycine or GABA receptors is one factor that leads to a local increase in gephyrin concentration, which drives E-domain dimerization, filament formation and synapse stabilization. In vitro, the E-domain can form filaments at high concentrations alone. However, in neurons, the G-domain is necessary for synapse formation. Therefore, it is believed that the G-domain fulfills gephyrin oligomerization at a higher level (Lardi-Studler et al., J Cell Sci, 2007, see next answer too).

4. Comparison to Hexagonal Lattice Model

The study dismisses the hexagonal lattice model (Saiyed et al., 2007) but does not directly contrast it with the filament model. A discussion of how these two models may (e.g., filaments as a dynamic precursor to lattices) or may not be mutually compatible would strengthen the paper.

ANSWER: We did not intend to dismiss the hexagonal lattice model and now discussed in the revised version our structural findings in light of this model.

Our data lends itself to the conclusion that a hexagonal lattice is not a requirement nor a precursor for gephyrin filaments, given that gephyrin filaments were also observed for isolated GephE dimers. However, based on studies from human patients with various neuropsychiatric disorders harboring gephyrin deletions in either GephG, GephC, or GephE (Linoel et al. HMG 2013) and in vitro studies (Lardi-Studler et al., J Cell Sci, 2007) it is established in the field that proper oligomerization of both terminal domains is required for synapse formation. Future studies are needed to structurally resolve the GephG in gephyrin full-length filaments. One possible role might be the generation of branching points providing GephE-dimerizations with other gephyrin trimers thus including elements of the hexagonal lattice model, while our current data, in agreement with existing electron tomographic data (Linsalata et al. J Comp Neurol 2014), excludes an ordered lattice as underlying structural principle of gephyrin clustering.

Minor Comments

5. Receptor Binding in the G375D Mutation

The authors reported that GephE-G375D was purified and analyzed by cryo-EM in the presence of a 5-fold molar excess of GlyR β -loop. However, Figure 5e (G375D dimer structure) shows no visible density for the GlyR β -loop, unlike the WT GephE-GlyR β -loop complex (Fig. 4). The absence of the receptor density in the mutant is puzzling and requires clarification.

ANSWER: The differences noted by reviewer #1 were due to different processing of the GephE G375D data compared to the wildtype GephE data. While focused classification was used to identify the subset of particles showing the best conserved GlyR β -loop density, we did not perform such processing for GephE G375D as previous studies by Dejanovic et al. EMBO Mol Med (2014) only observed a minor impact GlyR β -loop binding affinity in comparison to wildtype Gephyrin. When inspecting the data following focused classification, we also observed comparable density representing bound GlyR β -loop (see Figure R3 in this letter). Due to space constraints and the large set of supporting figures we have already, we would rather suggest to refrain from showing this data in the revised manuscript.

Figure 2: Workflow of GephEG375D to recover the density of GlyR-loop in the structure. The starting point was non-uniform refinement yielding the highest quality map (a, see sup-fig 13). Subsequently the particles were subjected to 3D classification with a focused mask covering the binding pockets on t GephE dimer (b). Two classes depicted additional density present at the binding pockets (green circles), whereas three classes showed no additional density (red circles). As exemplary particles and starting model from one 3D class showing no additional density (class 0) and one showing additional density (class 1) were further processed using non-uniform refinement. Both maps were low pass filtered to 6 Å (c, d). The resulting refinements resulted in two maps showing either additional density for the GlyR-loop present in the map (d, green density) or clearly the absent of such density (c) as has been seen previously in GephE WT example.

6. Specified the Molecular Weight of the Geph-GlyR Complex

The manuscript states that the Geph-GlyR β -loop complex elutes in SEC at the expected molecular weight of a trimeric complex (Sup-Fig 1a). However, a more rigorous molecular

weight determination would strengthen this claim, as SEC elution volume alone is not definitive due to shape and dynamic equilibrium of the complex.

ANSWER: We apologize if Sup-Fig 1a could be interpreted as observing the Geph-GlyR β -loop eluting at the expected molecular weight, which indeed is problematic for Gephyrin complexes due to exactly the reasons mentioned by reviewer #1. The complex eluted at the volume expected based on our experience working with recombinant *E. coli* expressed gephyrin for more than two decades. We have adapted the wording in our revised manuscript accordingly.

7. Statistical Reporting

Figure 5: Please specify the statistic test used (e.g., unpaired t-test, ANOVA?).

ANSWER: The statistic test used for the data shown in Figure 5 was a one-way ANOVA (Fig 5 d and i) and student t-test (Fig 5 b). We have added this to the Figure legend and added a short section to the Materials and Methods section.

8. Typo

Original problematic sentence: "Next, we asked whether the GephE filaments formation are to LLPS at inhibitory synapses, which has been hypothesized to represent the molecular basis of Geph clustering." A verb is missing between "are" and "to".

ANSWER: We have added the missing verb.

Reviewer #2 (Remarks to the Author):

In this manuscript, Macha and colleagues report the structures of Gephyrin E filaments using single-particle cryo-EM analysis. They examine the structures of Geph WT and Geph WT+GlyR at low concentrations (Fig. 1), Geph WT+GlyR at high concentrations where filaments were observed (Fig. 2a-b), GephE+GlyR filaments (Fig. 2c-e, 3-4), full-length Geph Δ SDII-GlyR, and GephE G375D/GephE D422N (Fig. 5). The structural study is well-executed and comprehensive. The observation of GephE filament formation and the interlocking interface of neighboring SDIIs is intriguing. Furthermore, the finding that SDII deletion abolishes filament formation, phase separation, and synaptic receptor clustering provides valuable insights into Gephyrin function. The analysis of disease-linked mutants further strengthens the study's significance.

We thank the reviewer for appreciating the importance of our structural work in understanding the synaptic function of gephyrin.

However, the physiological relevance of GephE filaments remains unclear. Additionally, the connection between filament formation, phase separation, and synaptic receptor clustering appears somewhat premature, as further evidence is needed to confirm Gephyrin's ability to undergo phase separation both in vitro and in neurons. Below are specific points for consideration:

1. As the authors note, GephE filaments form during cryo-EM sample preparation and can even assemble in the absence of GlyR. This raises questions regarding their physiological relevance and whether such filaments can form in a cellular environment.

ANSWER: We thank reviewer #2 for making us aware that our argumentation regarding the physiological relevance was not fully convincing, yet. Here, we would like to refer to our answer to comment 1 of reviewer #1, who addressed the same limitation, in this matter, and we believe we could clarify the concerns based on our studies using pathogenic gephyrin variants and further validation utilizing functional experiments in cultured cells.

2. The formation of filaments or LLPS typically requires multivalent interactions. For GephE, in the presence of GlyR, these interactions may be mediated by GlyR-induced dimerization and SDII interactions. In full-length Geph, additional multivalent factors, such as GephG trimerization, could facilitate assembly even in the absence of GlyR. Thus, the filament of the full-length Geph and Geph E might have different structures. Given that the authors successfully addressed preferred orientation in GephE filament analysis, could the same strategy be applied to solve the filament structure of full-length Geph, which may be more physiologically relevant?

ANSWER: We agree that GephG trimerization is also of strict importance for the formation of gephyrin-dependent post-synaptic density (Lardi-Studler et al. JCS 2007), see also our answer to comment 3 of reviewer #1). Our in vitro data shows similar filament formation for GephE and full-length gephyrin, while for the latter the crowdedness on the sample grid hindered high-resolution 3D reconstruction. Importantly, we clearly observe the same Z-shaped feature in 2D class averages obtained for the data of full-length gephyrin (Fig 2 a) and GephE (Fig 2 c), highlighting that the SDII-SDII' interface is not an artefact arising from the gephyrin truncation in the GephE construct. Tilting grids with full-length gephyrin, similar to our approach for GephE, did not yield data that would allow for a high-resolution reconstruction – indeed that was one reason for our decision to work on isolated GephE to obtain a high-resolution 3D structure of the SDII-SDII' interface.

3. In Figure 1e-g, is it possible to resolve GlyR in these structures? How does its structure compare to that of GlyR in the filament?

ANSWER: For clarification, we worked only with the cytosolic GlyR β -loop construct which binds to the dimeric gephyrin E domain. Assuming that the reviewer refers to this structure, we can only provide structural information within the complex with GephE either in the dimer or in the filaments. In the absence of gephyrin, the unbound free GlyR β -loop is too small to be observable by negative stain EM, which is limited in resolution by stain granularity.

4. On page 9, last line, should “GephE” be “GlyR”?

ANSWER: We assume the reviewer refers to this sentence:

„...we investigated if the insertion of GephE into filaments changes the position of the SDII in relation to the other domains of the protein.

This is correct, because we refer to the structure of GephE in the isolated dimer versus the filament and wanted to inspect potential structural changes on SDII. We changed the word “insertion” to “incorporation” to make it clear that we refer to the assembly of filaments.

5. The only evidence provided for Geph LLPS is a sedimentation assay. Filament formation is distinct from LLPS, which, by definition, involves liquid-liquid phase separation. Cryo-EM structural data on filaments do not constitute evidence for LLPS. To support LLPS, the liquid properties of the assembly should be demonstrated, for example, through imaging (e.g., round morphology), fusion events, or FRAP analysis.

ANSWER: We agree with the reviewer that filament formation per se is not LLPS. For gephyrin, the suggested experimental evidence has been reported in three recent landmark publications (Bai et al. Cell Res 2021, Lee et al. PNAS 2024, Zhu et al. Science 2025), demonstrating the typical physicochemical characteristics for LLPS by gephyrin. In our study, we now show that filaments formed by the GephE, the core element required for LLPS in gephyrin, may represent a structural feature of the specific still flexible and multimodal interactions of gephyrin in phase separated condensates. While we cannot say that GephE filaments are sufficient for LLPS in gephyrin, our mutational studies provide complementary evidence that in the absence of GephE filaments, LLPS (as shown here by sedimentation) is highly reduced.

6. Additional evidence could also strengthen the claim of Geph undergoing LLPS in cells. For instance, a FRAP experiment could assess the dynamics of Geph clusters in Figure 5A and H

ANSWER: We would like to refer to our answer to comment 5 above.

7. In Figure 5A and H, Geph foci colocalize with vGAT foci, which persist even in the presence of Geph Δ SDII, despite its inability to form foci. This raises the question of whether full-length Geph spontaneously forms foci via LLPS or is merely recruited into vGAT foci through vGAT binding.

ANSWER: We are unsure on how to interpret this comment, but do feel that we need to better explain the rationale of this experiment: In our cellular studies we use vGAT (vesicular GABA transporter) as a presynaptic marker to visualize active presynaptic sites that project to inhibitory synapses. There is no direct interaction or contact between the vGAT protein and gephyrin and to our knowledge, any lack of gephyrin will not change the ability of vGAT to localize to presynaptic sites. Therefore, the existence of vGAT positive presynaptic foci is not surprising even when Geph Δ SDII prevents formation of postsynaptic foci.

8. Could the reduced foci formation observed for GephE G375D/GephE D422N be due to

reduced vGAT binding rather than impaired assembly? Additionally, it would be helpful to confirm whether these mutants retain their ability to bind the GlyR loop, though this has been previously reported (references 31 and 33).

ANSWER: As stated above, to our knowledge there is no direct binding between Gephyrin and vGAT. We assume reviewer #2 means “reduced GlyR binding”. As stated by the reviewer, there is data showing that GlyR binding is retained in gephyrin G375D and gephyrin D422N with only moderately reduced affinity (Dejanovic et al. EMBO Mol Med 2015, Macha et al. HMG 2022, Bai et al., Cell Res 2021). While we fully agree that independent reproduction of such experiments is of advantage to the field, in this case the data came from our lab, and in case of gephyrin D422N even from the same first author.

9. The cryo-EM study on disease-linked mutants was conducted using the GephE domain alone. To better understand and compare their assembly properties, whether through filament formation or LLPS, it would be beneficial to perform sedimentation assays and/or cryo-EM analysis using full-length Geph with these mutations.

ANSWER: We agree with reviewer #2 that using the full-length protein for sedimentation assays and the cryo-EM analysis would be better, however we did not succeed in this as the full-length protein behaves more poorly compared to GephE. Most data we have on gephyrin is derived from the isolated GephE domain, and GephE has become an accepted model system for investigating gephyrin LLPS in the field (Bai et al. Cell Res 2021, Lee et al. PNAS 2024). With regard to the cryo-EM analysis of full-length gephyrin mutant variants we refer to our answer to comment 2 of reviewer #2: the SDII-SDII' interface appears identical for full-length gephyrin and GephE, at least on the level of high-resolution 2D class averages.

10. Currently, there is insufficient evidence to support the claim that Geph forms filaments or fiber-like structures in cells. Some statements should be revised accordingly, or additional experimental data should be provided. For example, on page 12, lines 2-3, the manuscript states: “Overall, these experiments demonstrate that the SDII domain-mediated filaments are a prerequisite for the formation of postsynaptic clusters at GABAergic synapses.” This conclusion would be more compelling with stronger supporting evidence.

ANSWER: We agree with the reviewer that we did not show the existence of those GephE filaments in cells directly. However, there are electron tomography studies of GABAergic synapses showing very similar filament-like structures (Linsalata et al. J Comp Neurol (2014). Furthermore, our in vitro and cell culture-based data demonstrate that SDII-mediated filament formation is required for synapse formation and that two pathogenic variants as well as a complementary charge variant all collectively fail to form filaments and synapses. In either case, we will reword as suggested by the reviewer to make clear that our statement is based on indirect but orthogonal experimental conclusions.

Minor wording:

11. The statement, “Next, we asked whether the GephE filaments formation are to LLPS at inhibitory synapses...” is unclear and should be revised for clarity.

12. In the Methods section, there is a heading for “Sedimentation and imaging assays of in vitro phase separation,” but no imaging assays are included. If imaging data were not performed, the heading should be revised accordingly.

13. In the Methods section, there is a typo in “Protein expression and purification” (should be “purification”).

14. In the Discussion, the manuscript states that for D422N, “receptor binding and Gephyrin’s enzymatic function remained largely unchanged,” citing reference 33. However, reference 33 (Macha et al.) appears to show that Geph D422N has reduced GlyR binding. This discrepancy should be clarified.

ANSWER: We thank the reviewer for pointing these wording issues out and we have corrected them in the revised manuscript.